# Fascin structural plasticity mediates flexible actin bundle construction

Rui Gong ●[1,4] ✉, Matthew J. Reynolds ●[1,4], Keith R. Carney ●[2,3], Keith Hamilton[1], Tamara C. Bidone[3] & Gregory M. Alushin ●[1] ✉

Fascin cross-links actin filaments (F-actin) into bundles that support tubular membrane protrusions including filopodia and stereocilia. Fascin dysregulation drives aberrant cell migration during metastasis, and fascin inhibitors are under development as cancer therapeutics. Here, we use cryo-EM, cryo-electron tomography coupled with custom denoising and computational modeling to probe human fascin-1's F-actin cross-linking mechanisms across spatial scales. Our fascin cross-bridge structure reveals an asymmetric F-actin binding conformation that is allosterically blocked by the inhibitor G2. Reconstructions of seven-filament hexagonal bundle elements, variability analysis and simulations show how structural plasticity enables fascin to bridge varied interfilament orientations, accommodating mismatches between F-actin's helical symmetry and bundle hexagonal packing. Tomography of many-filament bundles and modeling uncover geometric rules underlying emergent fascin binding patterns, as well as the accumulation of unfavorable cross-links that limit bundle size. Collectively, this work shows how fascin harnesses fine-tuned nanoscale structural dynamics to build and regulate micron-scale F-actin bundles.

Subcellular cytoskeletal networks are built by filament cross-linking proteins that specify diverse network geometries using poorly defined mechanisms[1–4]. Here, we focus on colinear actin filament (F-actin) assemblies with uniform polarity (hereafter referred to as 'bundles'), which are prominent in tubular membrane protrusions known as filopodia[5–7]. At the leading edge of migrating cells, filopodia function as dynamic antennae, which sense and respond to external cues that instruct cytoskeletal dynamics[7–12]. Physiologically, filopodia are necessary for axonal outgrowth and pathfinding, embryonic development and wound healing[7,10,11,13–18]. Pathologically, they are associated with enhanced migration of metastatic cancer cells[19–23]. Although the architecture, composition and signaling functions of filopodia have been extensively studied, the structural mechanisms mediating their assembly and regulation remain poorly understood.

The F-actin cross-linking protein fascin is critical for filopodia biogenesis[24,25]. It has at least two actin-binding sites (ABS) in the same polypeptide chain, enabling it to cross-link filaments as a monomer[26–29].

Fascin is a clinical biomarker for metastatic cancer, and cells overexpressing fascin feature an overabundance of filopodia-like protrusions that mediate enhanced tissue invasion and migration, which are correlated with poor prognosis[30–37]. Consequently, small-molecule fascin inhibitors have shown promise as cancer therapeutics in mouse models and are undergoing phase 2 clinical trials for the treatment of gynecological and breast cancers[38–40].

Fascin cross-linked F-actin bundles have long served as a model system for studying cytoskeletal network assembly principles[29,41–45]. Fascin bundles F-actin into hexagonal arrays in vitro and in filopodia[26,43,46–52]. Early studies of sea urchin fascin bundles suggested that the filaments are in perfect register, with identical rotational phases[43,47,49,50]. Each filament would feature three pairs of fascin cross-bridges evenly distributed across one helical crossover (approximately 13 actin subunits), oriented along the three diagonals of the hexagonal lattice. Fascin cross-bridges aligned along the same diagonals form linear transverse cross-bands perpendicular to the colinear filament axes (Fig. 1a–c)[43].

[1]Laboratory of Structural Biophysics and Mechanobiology, The Rockefeller University, New York, NY, USA. [2]Huntsman Cancer Institute, University of Utah, Salt Lake City, UT, USA. [3]Department of Biomedical Engineering, University of Utah, Salt Lake City, UT, USA. [4]These authors contributed equally: Rui Gong, Matthew J. Reynolds. ✉e-mail: rgong@rockefeller.edu; galushin@rockefeller.edu

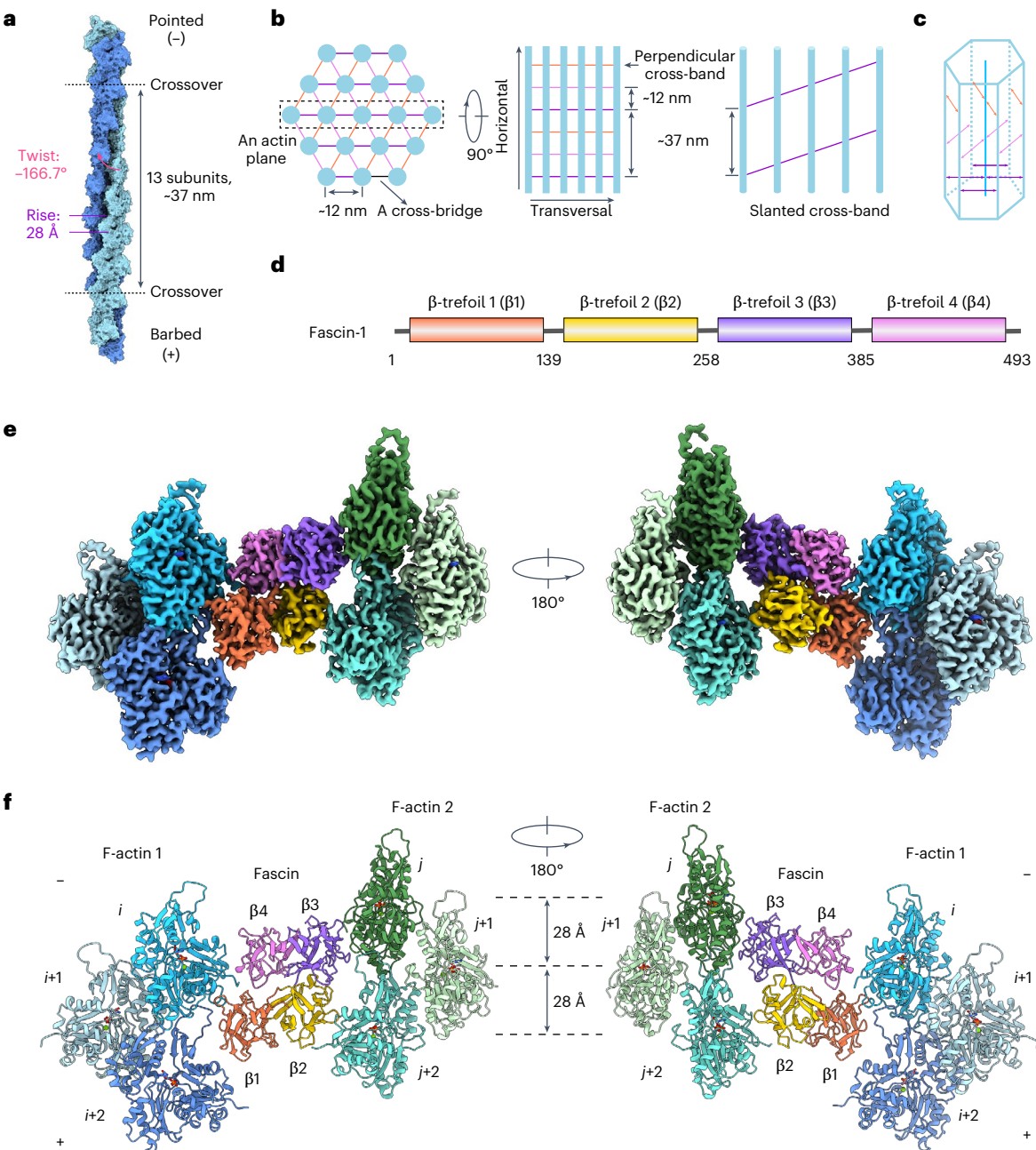

**Fig. 1 | Cryo-EM structure of fascin cross-linking actin filaments. a**, Diagram of F-actin's helical architecture. Model generated from PDB 7R8V. **b**, Left, schematic of end-on and side views of a fascin-cross-linked F-actin array. Actin filaments are shown in blue and fascin cross-bridges at each diagonal are depicted as colored lines. Right, side view of an actin plane (box in left panel) featuring a slanted fascin cross-band. **c**, Schematic of a fascin-cross-linked hexagonal bundle element. **d**, Domain architecture of human fascin-1. **e**, Composite 3.1-Å resolution cryo-EM density map of fascin-cross-linked F-actin. **f**, Atomic model of fascin-cross-linked F-actin in ribbon representation.

However, mismatches between F-actin's helical symmetry and the hexagonal array's translational symmetry render these bridging positions nonidentical[49]. More recent studies of vertebrate fascin bundles have instead suggested noncoherent filament rotational phases, with bundles featuring slanted or chevron-shaped cross-bands, implying a more complex cross-bridging pattern (Fig. 1b)[29,51–53]. Moreover, whereas arrays with perfect filament registration could theoretically grow indefinitely, bundles observed in filopodia or reconstituted in vitro feature approximately 20–30 laterally associated filaments[29,41,51,52], a size constraint that maintains filopodial mechanical compliance[42]. How fascin detects and modulates the architecture of F-actin bundles to facilitate filopodia size control have yet to be determined.

## Results

### Structure of fascin-bridging actin filaments

To gain insights into fascin's cross-linking mechanisms, we performed cryo-EM studies of F-actin bundles generated by human fascin-1 (Methods and Table 1). We initially obtained a consensus reconstruction containing one fascin molecule cross-linking two parallel actin filaments at 3.4 Å resolution (Extended Data Fig. 1a). However, this density map featured streaking artifacts indicative of conformational heterogeneity. We therefore performed two rounds of multibody refinement focused on each F-actin binding interface, yielding improved maps at 3.0 and 3.1 Å resolution, which were combined into a composite map in which both interfaces are well-resolved (Methods and Extended Data Fig. 1a–c).

**Table 1 | Cryo-EM data collection, refinement and validation statistics**

| | No. 1 Fascin cross-linked F-actin (Multibody: fascin bound filament 1) (EMDB-43364) (PDB 8VO5) | No. 2 Fascin cross-linked F-actin (Multibody: fascin bound filament 2) (EMDB-43365) (PDB 8VO6) | No. 3 Fascin cross-linked F-actin (Composite map) (EMDB-43366) (PDB 8VO7) | No. 4 Fascin cross-linked F-actin (Eigen_left) (EMDB-43367) (PDB 8VO8) | No. 5 Fascin cross-linked F-actin (Eigen_middle) (EMDB-43368) (PDB 8VO9) | No. 6 Fascin cross-linked F-actin (Eigen_right) (EMDB-43369) (PDB 8VOA) | No. 7 Bundle element, 460 Å (EMDB-43370) | No. 8 Bundle element, 740 Å (EMDB-43371) |
|---|---|---|---|---|---|---|---|---|
| **Data collection and processing** | | | | | | | | |
| Magnification | 29,000 | 29,000 | 29,000 | 29,000 | 29,000 | 29,000 | 29,000 | 29,000 |
| Voltage (kV) | 300 | 300 | 300 | 300 | 300 | 300 | 300 | 300 |
| Electron exposure (e⁻/Å²) | 61.26 | 61.26 | 61.26 | 61.26 | 61.26 | 61.26 | 61.26 | 61.26 |
| Defocus range (µm) | −0.8 to −2.2 | −0.8 to −2.2 | −0.8 to −2.2 | −0.8 to −2.2 | −0.8 to −2.2 | −0.8 to −2.2 | −0.8 to −2.2 | −0.8 to −2.2 |
| Pixel size (Å) | 1.03 | 1.03 | 1.03 | 1.03 | 1.03 | 1.03 | 1.03 | 1.03 |
| Symmetry imposed | C1 | C1 | C1 | C1 | C1 | C1 | C1 | C1 |
| Initial particle images (no.) | 3,056,360 | 3,056,360 | – | 113,800 | 113,800 | 113,800 | 3,056,360 | 3,056,360 |
| Final particle images (no.) | 113,800 | 113,800 | – | 17,207 | 79,824 | 16,769 | 8,477 | 8,053 |
| Map resolution (Å) | 3.0 | 3.1 | – | 3.9 | 3.4 | 4.0 | 8.7 | 12.0 |
| FSC threshold | 0.143 | 0.143 | | 0.143 | 0.143 | 0.143 | 0.143 | 0.143 |
| Map resolution range (Å) | 2.9–4.9 | 3.0–4.7 | – | 3.7–6.7 | 3.3–5.3 | 3.7–7.2 | 6.3–13.0 | 7.1–24.7 |
| **Refinement** | | | | | | | | |
| Initial model used (PDB code) | 7R8V, 3LLP | 7R8V, 3LLP | 7R8V, 3LLP | 7R8V, 3LLP | 7R8V, 3LLP | 7R8V, 3LLP | – | – |
| Model resolution (Å) | 3.1 | 3.0 | 3.1 | 3.6 | 3.5 | 3.6 | – | – |
| FSC threshold | 0.5 | 0.5 | 0.5 | 0.5 | 0.5 | 0.5 | | |
| Model resolution range (Å) | N.A. | N.A. | N.A. | N.A. | N.A. | N.A. | – | – |
| Map sharpening B factor (Å²) | −55.11 | −48.84 | – | −50.28 | −62.33 | −48.48 | −108.59 | −512.65 |
| Model composition | 3 actin protomers, 1 fascin | 3 actin protomers, 1 fascin | 6 actin protomers, 1 fascin | 6 actin protomers, 1 fascin | 6 actin protomers, 1 fascin | 6 actin protomers, 1 fascin | – | – |
| Non-hydrogen atoms | 12,625 | 12,625 | 21,460 | 21,460 | 21,460 | 21,460 | – | – |
| Protein residues | 1,604 | 1,604 | 2,723 | 2,723 | 2,723 | 2,723 | – | – |
| Ligands | 3 Mg.ADP | 3 Mg.ADP | 6 Mg.ADP | 6 Mg.ADP | 6 Mg.ADP | 6 Mg.ADP | – | – |
| **B factors (Å²)** | | | | | | | | |
| Protein | 23.94 | 27.87 | 43.13 | 99.43 | 78.72 | 112.12 | – | – |
| Ligand | 8.28 | 18.9 | 39.23 | 90.45 | 65.04 | 96.87 | – | – |
| **R.m.s. deviations** | | | | | | | | |
| Bond lengths (Å) | 0.003 | 0.005 | 0.004 | 0.004 | 0.002 | 0.004 | – | – |
| Bond angles (°) | 0.583 | 0.663 | 0.612 | 0.667 | 0.566 | 0.614 | – | – |
| **Validation** | | | | | | | | |
| MolProbity score | 1.55 | 1.73 | 1.55 | 1.94 | 1.63 | 2.07 | – | – |
| Clashscore | 7.37 | 8.65 | 8.07 | 16.31 | 8.24 | 18.66 | – | – |
| Poor rotamers (%) | 0 | 0 | 0 | 0.04 | 0.07 | 0.09 | – | – |
| **Ramachandran plot** | | | | | | | | |
| Favored (%) | 97.23 | 96.16 | 97.44 | 96.58 | 96.92 | 95.62 | – | – |
| Allowed (%) | 2.77 | 3.84 | 2.56 | 3.42 | 3.08 | 4.38 | – | – |
| Disallowed (%) | 0.00 | 0.00 | 0.00 | 0.00 | 0.00 | 0.00 | – | – |

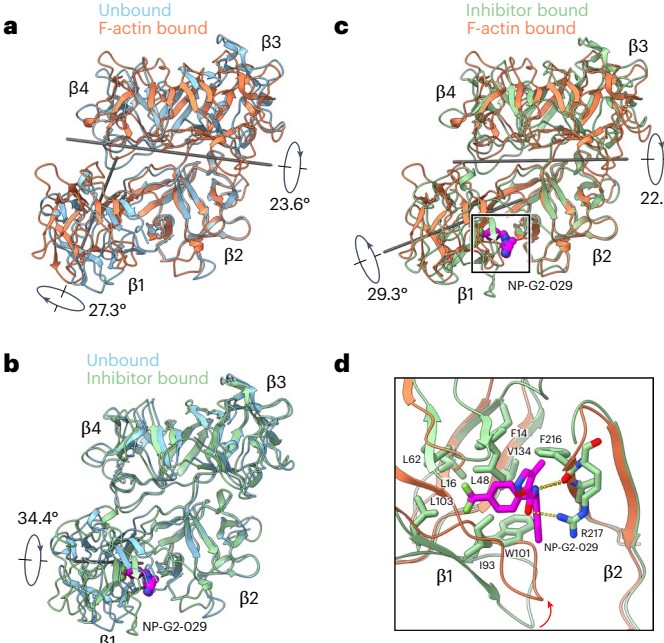

**Fig. 2 | Fascin rearrangements upon F-actin binding are allosterically blocked by G2. a**, Superposition of fascin in the prebound (PDB 3LLP) and F-actin bound states. **b**, Superposition of fascin in the prebound and inhibitor-bound (PDB 6B0T) states. The inhibitor NP-G2-029 is highlighted in space-filling representation. **c**, Superposition of fascin in the inhibitor-bound and F-actin bound states. **d**, Detail view of boxed region in **c** highlighting clash that would occur between NP-G2-029 and β-trefoil 1 in the F-actin bound state. All structures are aligned on β-trefoil 2.

In our structure, fascin uses two distinct ABS to cross-link a pair of actin filaments with an interfilament distance of ~12 nm. Fascin's overall shape is similar to that of the protein in isolation[27–29], with its four β-trefoil domains arranged in a compact, bent horseshoe-shaped conformation (Fig. 1d–f). It consists of two pseudo-twofold symmetry-related lobes: one lobe comprising β-trefoils 1 and 2 and the other β-trefoils 3 and 4. The two ABS are located on opposing surfaces at the interfaces between these lobes, with β-trefoils 1 and 4 forming ABS1, and β-trefoils 2 and 3 forming ABS2 (Fig. 1f).

**Fascin rearrangements upon binding F-actin are blocked by G2**
When compared with fascin in isolation (PDB 3LLP, which we refer to as the 'prebound' state)[27], the conformation of each β-trefoil domain is nearly identical after F-actin binding (Extended Data Fig. 2a). However, substantial interdomain rearrangements occur. The β-trefoil 1–2 lobe rotates 23.6° towards the β-trefoil 3–4 lobe, facilitated by the flexible β-trefoil 2–3 linker (residues 256–261). In addition, β-trefoil 1 undergoes an independent 27.3° rotation via a hinge region (residues 137–141) connecting β-trefoils 1 and 2 (Fig. 2a and Supplementary Video 1). These interdomain rotations substantially remodel fascin's surface, sculpting the two ABS into actin-binding competent conformations.

We next sought to interpret the mechanisms of fascin inhibition by the small molecule NP-G2-029 (G2). Previous structural studies of isolated fascin bound to G2 (PDB 6B0T) showed a substantial 34.4° rotation of β-trefoil 1 versus the prebound state[54], suggesting that G2 might disrupt ABS formation (Fig. 2b and Supplementary Video 1). Consistently, β-trefoil 1 undergoes rotations with distinct directions and magnitudes when the G2-bound structure is compared with the prebound state versus when our F-actin bridging structure is compared with the prebound state. The orientation of β-trefoil 1 differs between the actin-bridging and G2-bound structures by 29.3°,

resulting in disruption of ABS1 in the G2-bound structure (Fig. 2c,d and Supplementary Video 1).

**Fascin's F-actin binding interfaces**
Like many F-actin binding proteins, both fascin ABS engage a cleft formed by two longitudinally adjacent actin subunits, composed of each subunit's subdomain 1 and the D-loop of subunit $i + 2$ or $j + 2$ (Figs. 1f and 3a). At the fascin binding site, these two identical clefts on F-actin 1 and F-actin 2 are facing inwards towards the cross-bridge. Fascin's bent horseshoe shape dovetails with this bi-partite, doubly-spiraling interface, enabling each ABS to form extensive contacts that enforce precise bundle geometry. Because F-actin 1 and F-actin 2 are superimposable, both fascin ABS can engage the opposite filament by reorientating the molecule (Fig. 1f), producing two fundamental binding poses, an 'up' orientation (Fig. 1f, left) and a 'down' orientation (Fig. 1f, right).

Although all of fascin's β-trefoil domains contact the same F-actin cleft, the specific interactions are distinct, resulting in asymmetry between the two interfaces (Fig. 3a). β-Trefoil 1 makes extensive contacts with two separate patches on F-actin 1. Towards the plus ('barbed') end, fascin K74, D75 and F122 engage with R95 of subunit $i + 2$ (Fig. 3b). K74 and/or D75 or F122 mutations were previously reported to have no effect on fascin's bundling activity[26], suggesting this interface is functionally nonessential. Towards the minus ('pointed') end, β-trefoil 1 tightly packs against a large surface formed by subunit $i + 2$'s D-loop and subunit $i$'s subdomain 1 (Fig. 3c). Fascin F29, which is essential for bundling[26], is deeply buried in the hydrophobic groove at the intersubunit longitudinal interface. Fascin S39, a protein kinase C phosphorylation site that impairs F-actin binding and bundling[55], forms a potential hydrogen bond with the N-terminal residue E4 on subunit $i$. Phosphorylating S39 would electrostatically repel actin's negatively charged N terminus, thereby suppressing F-actin engagement. Several other fascin residues critical for F-actin bundling, including E27 and K43 (ref. 26), also mediate interactions at this interface. β-Trefoil 4 exclusively interacts with subunit $i$'s subdomain 1 through hydrophobic and electrostatic interactions (Fig. 3d). Notably, fascin R398, which is essential for bundling activity[26], forms a salt bridge with actin E100.

β-Trefoil 2 solely interacts with subdomain 1 of F-actin 2 subunit $j + 2$ (Fig. 3e). Fascin R151 and D168 form salt bridges with actin E93 and R95, respectively. R151 and D168 mutations have no effect on bundling activity[26], indicating these interactions are auxiliary. Notably, fascin R149 and actin R28 form an electrostatics-defying $\pi$-cation/$\pi$-cation interaction through cation clustering[56]. Mutation of R149 severely impairs fascin bundling activity[26], suggesting this unexpected interaction plays a major role in β-trefoil 2's actin engagement. β-Trefoil 3 forms an extensive interface with F-actin 2 subunits $j$ and $j + 2$, primarily through electrostatic interactions and Van der Waals contacts with subunit $j$'s subdomain 1 (Fig. 3f). In summary, we find that ABS1 and ABS2 form completely divergent contacts with F-actin despite fascin's pseudo-twofold symmetry.

**Bundle elements feature variable cross-linking architecture**
We next sought to determine how fascin cross-links helical F-actin filaments into hexagonal arrays despite their incompatible symmetries. We focused our analysis on 'bundle elements', minimal hexagonal sets of one central and six peripheral fascin cross-linked filaments. We obtained an 8.7-Å reconstruction of one bundle element class (representing only 0.3% of the initial picks) spanning one F-actin crossover length, which features substantial fascin occupancy at all bridgeable positions (Fig. 4a and Extended Data Fig. 2b,c). The overall organization of this bundle element is similar to that inferred from previous lower-resolution tomographic studies[26,43,51]. The seven filaments on the hexagonal lattice's vertices produce 12 equally spaced filament pairs separated by ~12 nm, each cross-linked by a single fascin. As anticipated, the 12 fascins are aligned along the 3 diagonals of the hexagon

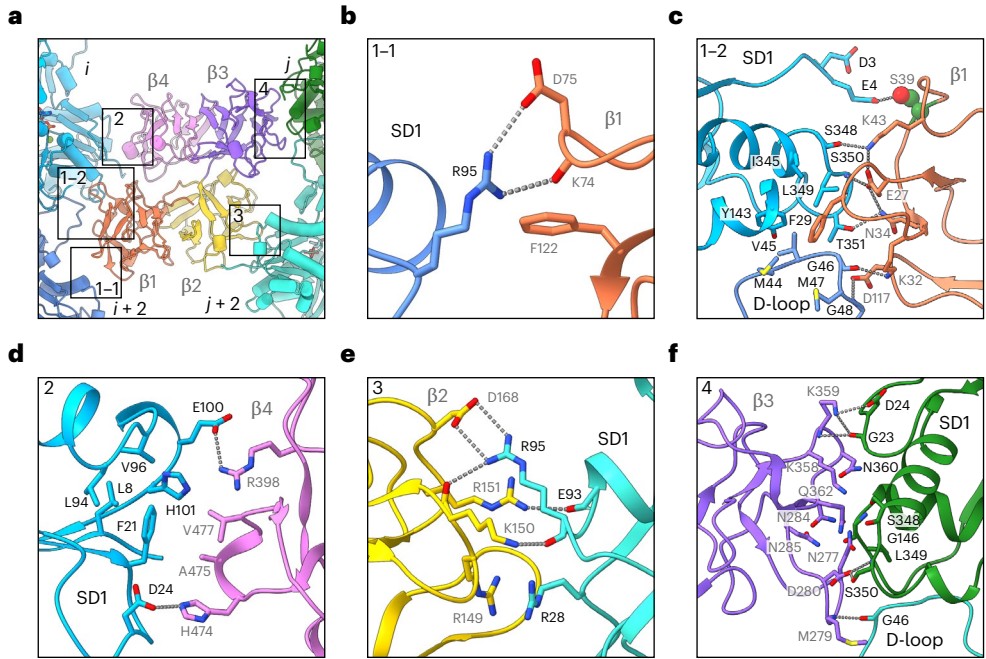

**Fig. 3 | Fascin's two F-actin binding interfaces are chemically distinct.**
**a**, Overview of fascin's two F-actin binding interfaces. **b**, Contacts between
subdomain 1 (SD1) of actin subunit *i* + 2 and fascin β1. **c**, Contacts between SD1 of
actin subunit *i*, D-loop of subunit *i* + 2 and fascin β1. Protein kinase C target residue
S39 of fascin is highlighted in space-filling representation. **d**, Contacts between SD1
of actin subunit *i* and fascin β4. **e**, Contacts between SD1 of actin subunit *j* + 2 and
fascin β2, including cation clustering interaction between fascin R149 and actin R28.
**f**, Contacts between SD1 of actin subunit *j*, D-loop of subunit *j* + 2 and fascin β3.

(Fig. 4a,b). Each set of four fascin cross-bridges oriented along the same
diagonal (different colors in Fig. 4a) occupies an approximate plane
perpendicular to the aligned filament axes (a transversal layer), and
the three transversal layers are roughly evenly spaced along the span
of the crossover (Fig. 4a,b). However, our subnanometer-resolution
reconstruction reveals that the fascins comprising each layer exhibit
orientational and positional heterogeneity.

Each transversal layer features fascin cross-bridges in both the 'up'
and 'down' poses (Fig. 1f), which are irregularly distributed throughout
the bundle element (Fig. 4b, Extended Data Fig. 2e,f and Supplementary
Video 2). For example, in the actin plane spanning filaments 6–0–3,
fascin c is 'up', whereas fascin d is 'down' (Fig. 4b, lower). In addition,
within a transversal layer, fascins featuring the same pose can be dis-
placed along the longitudinal axis. For example, in the actin plane
spanning filaments 1–0–4, the two 'up' fascins e and f are offset by
one actin subunit (Fig. 4c and Extended Data Fig. 2e). The two fascin
cross-linking poses and axial shifts in fascin positioning together pro-
duce the noncoherent arrangement of cross-bridges across transversal
layers. Although our bundle element reconstruction reveals substan-
tial lateral disorder, F-actin's helicity could nevertheless give rise to
longitudinally repetitive binding patterns. To probe this periodicity,
we obtained a 12.0-Å reconstruction of the bundle element using a
larger box spanning approximately two F-actin crossovers (Extended
Data Fig. 2b,d). The pose and position of each fascin precisely repeat
in the next crossover (Fig. 4d and Supplementary Video 2), consistent
with previous in vitro and in situ studies[26,29,43,51,52]. Collectively, we find
that flexibility in fascin's binding pose and axial positioning allows the
protein to accommodate a range of interfilament geometries encoun-
tered in bundles.

**Fascin cross-linked filaments are rotationally noncoherent**
Early theory predicted the emergence of linear cross-bands when
bundled actin filaments are in register[43,49]. Instead, we observed only
partially ordered transversal layers featuring axially shifted fascins in
our bundle element reconstruction. Because the filaments exhibited

minimal axial translations (0.3–2.1 Å; Extended Data Table 1), we
hypothesized that they were instead rotationally noncoherent. We
therefore measured the rotational phase shift of the six peripheral fila-
ments versus the central filament, which we define as the axial rotation
required to superimpose a filament with the central reference when
viewed from the minus end. We found that all six peripheral filaments
indeed exhibit distinct rotational phase shifts that vary in direction and
magnitude, which broadly fall into two groups (Fig. 4e).

One group, consisting of filaments 2, 3 and 6, displays minor phase
shifts of 5.8°, −4.1° and −2°, respectively. Because these values approxi-
mate 0°, the filaments in the 6–0–3 actin plane are nearly in register,
with highly similar interfaces between filament pairs 6–0 and 0–3
(Fig. 4b, lower). However, fascin c, cross-linking filament pair 6–0,
adopts the 'up' pose, whereas fascin d, cross-linking filament pair 0–3,
adopts the 'down' pose. This suggests that pairs of filaments that are
rotationally in-phase can nevertheless be cross-linked by either binding
mode, a degeneracy anticipated to produce cross-band disorder even
in well-registered filament arrays. The other group includes filaments 1,
4 and 5, which display substantial rotational phase shifts of −23.1°, 23.2°
and 20.9°, respectively. Given the helical symmetry of F-actin, a rotation
of ±26.7° is equivalent to an axial translation of one actin subunit. In the
1–0–4 actin plane, two such consecutive rotations produce equivalent
fascin cross-linking sites axially shifted by one actin subunit. Fascins e
and f bind at these sites, both in the 'up' pose, resulting in a prototype
of a slanted cross-band (Fig. 4c and Extended Data Fig. 2e).

Because fascin's cross-linking geometry varies with the rotational
phase shifts between filaments, we hypothesized that other bundle ele-
ments with alternative interfilament rotation and/or cross-linking pat-
terns could be present. We therefore examined four additional bundle
element classes in which the central actin filament and its six associated
fascin cross-bridges were resolved (Extended Data Fig. 2b). When these
four models are superimposed with the first bundle element, all five
models feature three pairs of fascin cross-bridges clustered at six dis-
tinct locations consistent with three transversal layers (Extended Data
Fig. 2g–i). At each location, both the 'up' and 'down' cross-linking poses

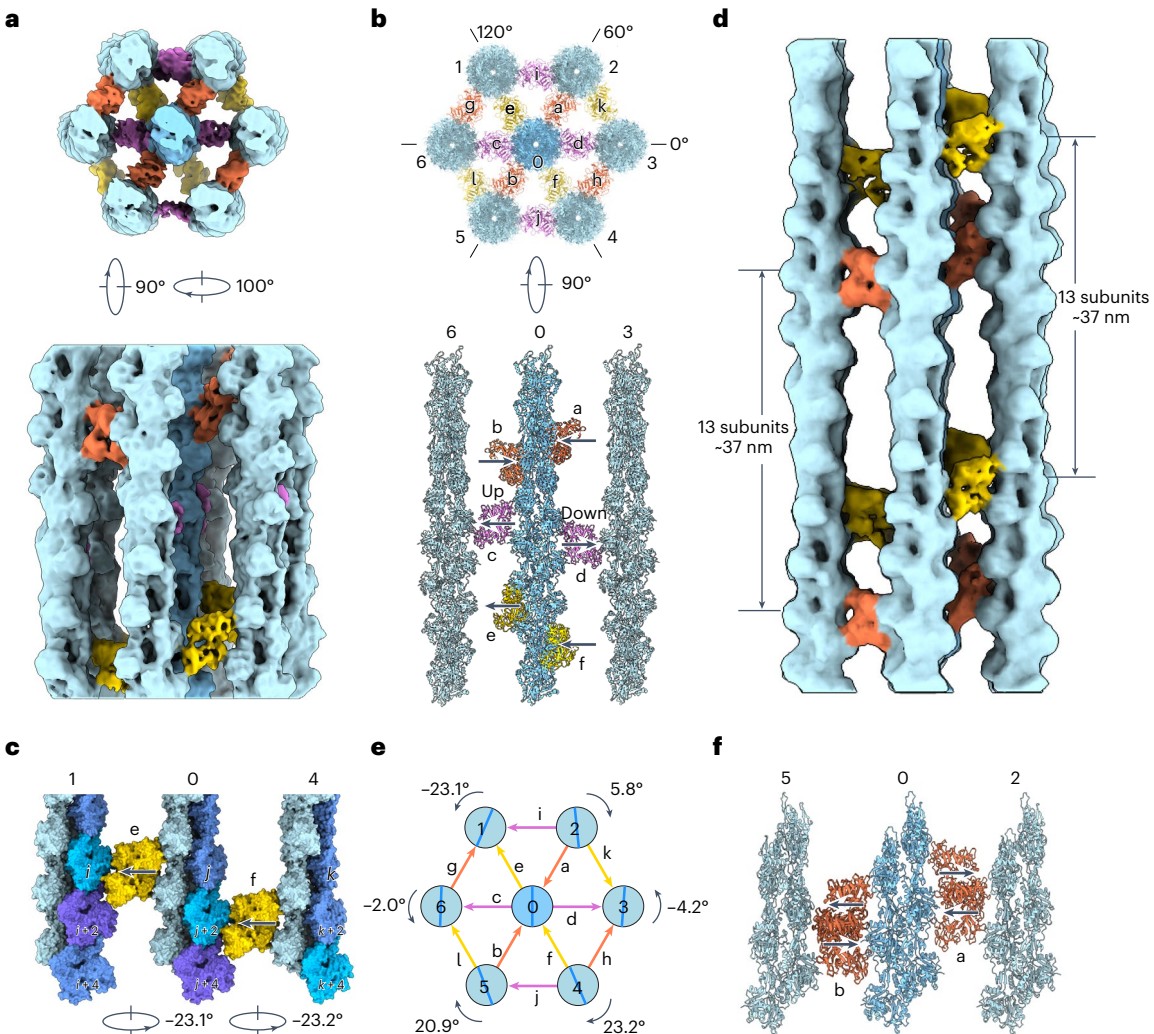

**Fig. 4 | Architecture of a fascin cross-linked F-actin hexagonal bundle element. a**, End-on (upper) and side view (lower) of 8.7-Å resolution fascin cross-linked F-actin bundle element reconstruction (box size: 460 Å). Actin filaments are colored in shades of blue and fascins comprising each transversal layer are colored in different hues. **b**, End-on (upper) and side view (actin plane 6–0–3; lower) of bundle element docking model in ribbon representation. The 'up' and 'down' poses of fascin c and d are indicated by arrows, with the arrowhead pointing towards ABS1. This orientation designation is used throughout the remainder of the paper. **c**, Side view of actin plane 1–0–4, highlighting axial shift of fascin cross-bridges associated with sequential filament rotational phase shifts. **d**, Side view of 12.0-Å resolution bundle element reconstruction from particles re-extracted with a box size of 740 Å. **e**, Diagram of peripheral filament rotational phase shifts relative to the central filament, as well as fascin cross-bridge poses. **f**, Side view (actin plane 5–0–2) of five bundle element docking models superimposed on their central filaments, highlighting variations in fascin pose and axial positioning at equivalent bridging locations.

occur with varying frequencies (Fig. 4f and Extended Data Fig. 2g–i). None of the classes share the same distribution of fascin poses, suggesting that each class represents a unique bundle element geometry (Fig. 4e and Extended Data Fig. 2j). We furthermore observed that for a given transversal layer, different classes featured pairs of fascins with the same pose that were either colinear or axially shifted by one actin subunit, suggesting varying filament rotational phases. In addition, the occupancy of individual fascins and filaments varies substantially between bundle element classes (Extended Data Fig. 2b), implying differential stability.

A previous study suggested that altering F-actin's helical twist contributes to hexagonal fascin bundle formation[41]. Substantial twist changes were also reported in early nanometer-resolution cryo-EM studies of acrosomal bundles cross-linked by scruin, which was also suggested to remodel actin protomers[57]. To assess this potential additional mechanism, we measured the filaments' helical parameters in our subnanometer-resolution bundle element reconstruction (Fig. 4a). Analysis with two software packages (Methods) showed that all the filaments had a nearly identical consensus twist (Extended Data Table 1), with both packages producing similar measurements (−166.6 ± 0.4° versus −166.4 ± 0.2°). These values are highly similar to canonical F-actin's twist of −166.7° (ref. 58). To assess potential asymmetric twist variations, as reported for scruin bundles[57], we also measured local twist along the filaments[58]. We found minimal internal differences in each filament (Extended Data Table 1), with subtle variations similar to those previously reported in an subnanometer-resolution asymmetric reconstruction of undecorated ADP F-actin[58]. Collectively, this analysis suggests that under our experimental conditions, fascin does not substantially alter F-actin's helical twist.

Because F-actin's helix undergoes slightly more than six full turns in the 13-subunit crossover length, each crossover layer will be subtly rotated by approximately 3–5° relative to its axial neighbors. However, because of F-actin's helical symmetry, the interfilament rotational phase shifts effectively remain constant, thereby preserving transversal layer internal geometry. Nevertheless, variability in F-actin's helical twist may erode this longitudinal order at longer length scales[59,60].

Regardless, our data suggest that fascin's two binding poses coupled with interfilament rotational freedom are sufficient for fascin to assemble hexagonal bundle elements with varying geometries, without substantially remodeling F-actin's helix.

### Fascin structural plasticity enables flexible cross-linking

Fascin's capacity to accommodate varying binding positions has been predicted to require cross-bridge structural plasticity[49]. The major variability mode in our multibody refinement features a continuous counterclockwise rotation of F-actin 2 relative to F-actin 1 when viewed from the minus end, while their filament axes remain parallel. To assess the structural underpinnings of this variation, particles were partitioned into three groups along the corresponding eigenvector (Fig. 5a). Subsequent refinements resulted in three reconstructions at resolutions of 3.9, 3.4 and 4.0 Å, representing snapshots along the conformational continuum that we refer to as eigen_left, eigen_middle, and eigen_right (Fig. 5b and Extended Data Fig. 3a–c).

Both actin filaments feature indistinguishable conformations across all three snapshots, indicating that F-actin structural plasticity does not substantially contribute to cross-bridge flexibility (Extended Data Fig. 3d). Comparing the eigen_left and eigen_middle snapshots shows no distinguishable changes at either of the ABS–F-actin interfaces, suggesting fascin rearrangements are primarily responsible for this aspect of the conformational landscape (Fig. 5c,d and Extended Data Fig. 3e–k). Consistently, in the reference frame of F-actin 1, β-trefoil 1 remains stationary, whereas β-trefoils 2–4 undergo a rigid-body rotation through two pivot points: the F-actin 1–β-trefoil 4 interface and the flexible linker connecting β-trefoils 1 and 2 (residues 137–141) (Fig. 5c and Supplementary Video 3). Notably, the actin-binding residues on β-trefoil 4 are clustered in two pliant surface loops (Fig. 3d and Extended Data Fig. 3h), allowing β-trefoil 4 to rotate while maintaining interactions with the filament.

Conversely, when the eigen_middle and eigen_right snapshots are compared in the reference frame of F-actin 1, fascin's conformation remains essentially unchanged, as does the ABS1–F-actin 1 interface (Fig. 5b,e and Extended Data Fig. 3e–h). However, alignment on F-actin 2 reveals substantial remodeling of the ABS2–F-actin 2 interface, with both β-trefoils 2 and 3 displaced relative to the filament surface (Fig. 5f). At the β-trefoil 2–actin interface, key interactions are maintained by repositioning the side chains of residues located on flexible loops (Extended Data Fig. 3l). However, at the actin–β-trefoil 3 interface, both actin-interacting loops on β-trefoil 3 are displaced, suggesting this contact is destabilized in the eigen_right snapshot (Extended Data Fig. 3m). Taken together, we find that remodeling of both fascin and the ABS2–F-actin interface mediate cross-bridge structural plasticity (Supplementary Video 3).

We next measured the interfilament rotation angles in the snapshots, which reflect the distribution of rotational phase shift magnitudes in the bundle elements from which the particles were extracted. With F-actin 1 as the reference, the eigen_left, eigen_middle and eigen_right snapshots feature offsets of 9.1°, 20.1° and 30.7°, respectively (Fig. 5a), supporting substantial rotational heterogeneity in fascin bundles. Because the eigen_middle reconstruction comprises particles at the center of the distribution, interfilament rotational phase shifts approximately 20° in magnitude appear to maximize cross-bridge favorability.

### Modeling recapitulates variable bundle element architecture

To assess whether the notable structural features we identified are the major determinants of fascin bundle construction, we developed a minimal computational model. In our model, actin filaments are sequentially added to a bundle with rotational phase shifts that maximize the probability of fascin cross-linking. Fascin's binding probability at each potential bridging position between a newly added filament and pre-existing filaments is assessed through a scoring function parameterized by our multibody refinement analysis. Positions with scores above an empirical cutoff are initially nominated as candidates for cross-bridging. From these candidate positions, the rotational shift corresponding to the maximum overall score is selected, and the corresponding protomers are assigned as having fascin bound (Methods, Extended Data Fig. 4 and Extended Data Table 2).

We generated seven-filament assemblies corresponding to bundle elements. Because of the finite lengths of the filaments being simulated, we found that the orientation of the central filament relative to the simulation reference frame impacted the fascin binding patterns (Methods). We therefore generated assemblies featuring central filaments with absolute rotational shifts between 0° and 10° for detailed analysis. Consistent with our experimental reconstructions, the simulated assemblies feature substantial variability in the rotational phases of their component filaments and in the positioning of individual fascins, while nevertheless maintaining overall transversal layer organization (Fig. 5g). This is visualized in an averaged contact map (Fig. 5h), which consists of sparse peaks corresponding to allowable bridging locations that are broadened by axial positioning variability. A contact map calculated from the experimental bundle element classes is similar, particularly for cross-bridges contacting the central filament (Fig. 5h). Discrepancies in outer filament cross-bridging patterns are likely caused by the reductionist nature of the model, as well as potential nonrepresentative sampling by our three-dimensional (3D) classification procedure, which only recovers averageable classes. Nevertheless, the general correspondence between the model and our experimental data suggests that the model's features, which minimally encode fascin-binding probability modulated by interfilament rotation angles, are largely sufficient to recapitulate the observed variable internal bundle geometry.

### Filament rotations establish emergent cross-band patterns

To examine whether the structural plasticity of bundle elements is linked to emergent fascin binding patterns, we analyzed reconstituted bundles using cryo-electron tomography (cryo-ET) (Methods and Extended Data Table 3). Cryo-ET's potential to produce molecular-resolution 3D reconstructions has been practically limited by low signal-to-noise and nonisotropic resolution. To overcome these limitations, we adapted our neural network-based processing framework to denoise our tomograms and semantically segment fascin and F-actin (Methods and Extended Data Fig. 5). The unprecedented molecular details revealed by this procedure allowed us to unambiguously distinguish the 3D positions and orientations of actin filaments and individual 50 kDa fascin molecules without averaging (Fig. 6a and Supplementary Video 4), facilitating direct analysis of bundle network geometry.

Bundles showed nearly complete fascin decoration, with the ~37 nm axial periodicity of cross-bands persisting for tens of crossovers across the entire region visualized for a straight bundle (Fig. 6a). Bundles featured an oblong aspect ratio when viewed end-on. They were typically three to five layers thick along the z direction but could be more than ten layers wide. Many bundles had substantial morphological flexibility and spatially varying filament composition at the micrometer scale, including filament splaying, interweaving between bundles and bundle coalescence (Extended Data Fig. 6a,b), as has been observed in filopodia[51,52]. Moreover, we observed examples of bundles whose filaments collectively twist along the longitudinal axis (Extended Data Fig. 6b).

To study the relationship between bundle geometry and molecular organization, we selected four bundles composed of 29–41 straight filaments for detailed analysis. We were able to rigid-body dock atomic models of fascin cross-bridges and actin filaments directly into the denoised tomograms, allowing us to assign fascin positions and poses while measuring filament rotational phase shifts (Fig. 6b and Extended Data Fig. 6c,d). We occasionally observed actin planes in which all filaments are consecutively rotated by approximately −25° (Extended

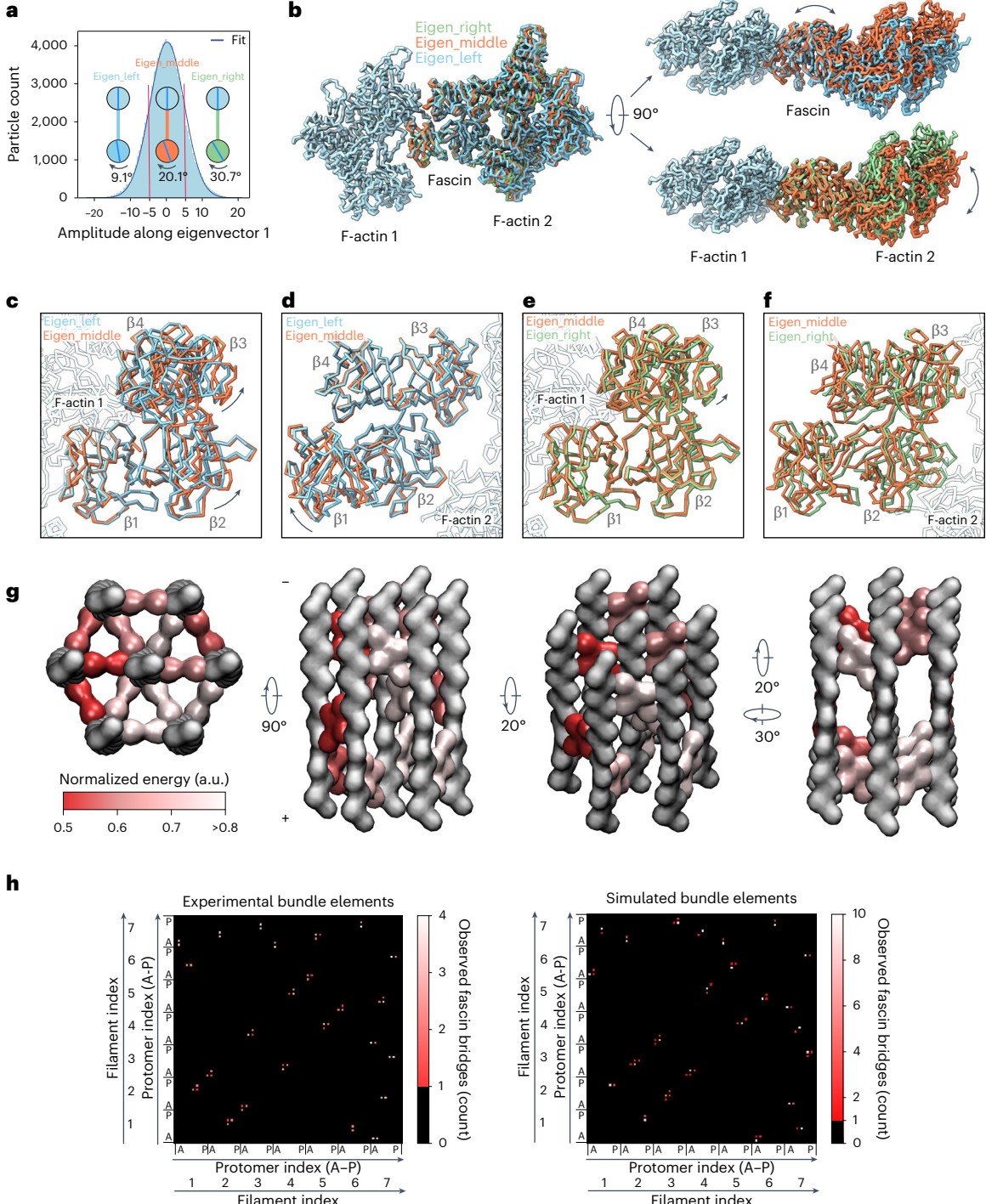

**Fig. 5 | Fascin cross-bridge structural flexibility mediates interfilament rotations. a**, Histogram of amplitudes along the first multibody refinement eigenvector for all particles ($n$ = 113,800), which follows a unimodal Gaussian distribution ($R^2$ = 0.9986). Red lines indicate cutoffs for partitioning particles into three bins. Rotational phase differences between filaments from each reconstructed bin are depicted. **b**, Side (left) and end-on (right) views of atomic models from eigenvalue binned reconstructions superimposed on F-actin 1. Predominant structural transitions explaining the relative rotation of F-actin 2 are indicated by double-headed arrows. **c,d**, Structural comparison of eigen_left and eigen_middle atomic models when superimposed either on F-actin 1 (**c**) or F-actin 2 (**d**). Rigid-body repositioning of fascin subdomains are indicated

by black arrows. **e,f**, Structural comparison of eigen_middle and eigen_right reconstructions when superimposed either on F-actin 1 (**e**) or F-actin 2 (**f**). **g**, Views of a representative simulated hexagonal bundle element. Actin is shown in gray and fascins are colored by normalized energy scores. **h**, Contact maps of experimental ($n$ = 5) and simulated ($n$ = 10) hexagonal bundle elements. Models are aligned on the central filament. Each filament contains 16 protomers (A–P) indexed alphabetically from the barbed end to the pointed end. The peripheral filaments are numbered sequentially from 1 to 6 (as arranged in Fig. 4b), whereas the central filament is assigned index 7. Each matrix element depicts a potential fascin cross-linker position in the bundle, whereas color corresponds to the observed number of fascin cross-bridges. a.u., arbitrary units.

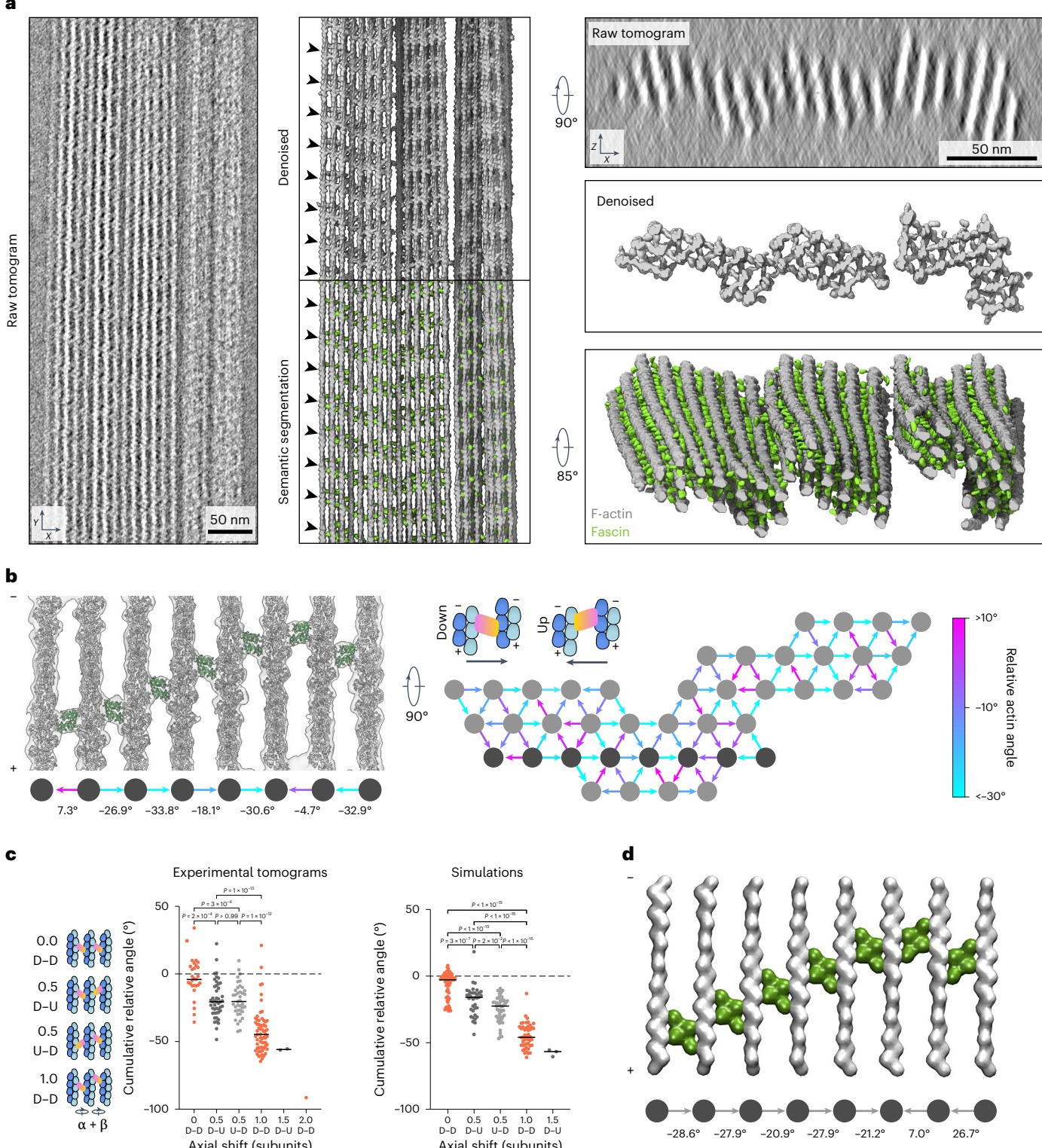

**Fig. 6 | Denoised tomograms reveal nanoscale organization of fascin-cross-linked bundles. a**, Left, side views of projected raw, denoised (middle upper) and semantically segmented (middle lower) tomogram. Arrowheads indicate the periodic chevron-like pattern of fascin cross-bridges. Right, end-on views of projected raw (upper) and denoised (middle) tomogram, as well as an oblique view of semantic segmentation (lower). **b**, Left, side view of representative rigid-body docking model of one actin plane from a denoised tomogram. Filament rotational phase shifts and fascin poses (arrows) are indicated. Right, end-on view schematic of filament rotational phase shifts and fascin poses throughout a 17-protomer slab of a bundle. Dark gray filaments correspond to the actin plane

shown on the left. **c**, Quantification of cumulative filament rotational phase shifts versus fascin axial shifts across observed three-filament configurations from denoised tomograms ($n = 293$ from $N = 5$ bundles) and simulations ($n = 202$ from $N = 8$ bundles). Orange points indicate matched fascin poses, whereas points in shades of gray indicate mismatched fascin poses. Data were compared by one-way analysis of variance using the Tukey method to correct for multiple comparisons. **d**, Side view of the central actin plane from a representative simulation of a 22-filament bundle. Filament rotational phase shifts and fascin poses (arrows) are indicated. D, down; U, up.

Data Fig. 6d). Correspondingly, the fascins are all in the 'down' pose and sequentially axially shifted to form a slanted cross-band. More frequently, cross-bands adopt a chevron morphology through a combination of consecutive approximately −25° rotations followed by a series of approximately +25° rotations (Fig. 6b). Despite this short-range order, globally we observe extensive heterogeneity in bundle element architecture (Fig. 6b and Extended Data Fig. 6c), consistent with our 3D classification analysis (Fig. 4e and Extended Data Fig. 2b,j).

We next examined how fascin binding patterns are propagated across bundles. We analyzed coplanar sets of three actin filaments featuring two fascin cross-bridges, minimal units that connect adjacent, partially overlapping bundle elements. After standardizing the viewing orientation such that only unique views were retained, we identified rules relating the cumulative rotational phase shift across the three filaments to patterns of fascin binding poses and axial offsets that produce chevron-shaped cross-bands (Fig. 6c). If the cumulative rotational phase shift is approximately 0°, both fascins will be in the same orientation with no axial shift. If the cumulative rotational phase shift is approximately −25°, the fascins will be oppositely oriented and have an axial shift of half an actin subunit length, with no pose preference for the first fascin. If the cumulative rotational phase shift is doubled to approximately −50°, the fascins will be axially offset by one actin subunit length and both be in the same ('down') pose. Because these cumulative rotational phase shifts approximate multiples of the rotation corresponding to a one subunit shift along F-actin (±26.7°), these patterns likely emerge to accommodate both F-actin's helical symmetry and plausible cross-linking positions in the hexagonal bundle lattice. Rarely, more extreme axial shifts were observed (1.5 or 2 subunit lengths), which followed the trend of oppositely oriented fascins for half-subunit length shifts and equivalently oriented fascins for full subunit length shifts.

Notably, there is some overlap in the cumulative phase shifts between configurations featuring distinct fascin axial positionings (Fig. 6c), consistent with fascin's capacity to adopt either the 'up' or 'down' pose when the rotational phase shift magnitude between two filaments is low. This orientational flexibility introduces wobble positions that facilitate chevron formation by switching the direction of slanted cross-bands. This likely contributes to bundle stability by allowing fascin to tolerate diverse binding site geometries and corresponding interfilament architectures.

We next assessed whether similar binding patterns can emerge in our computational model by simulating bundles of 22 filaments arranged into three layers (Methods). Indeed, we found substantial enrichment of slanted and chevron-shaped cross-bands, visible in both individual assemblies (Fig. 6d) and in fascin positioning across trials, which reproduced the experimentally observed binding patterns (Fig. 6c). This suggests geometric linkages between binding sites produce long-range patterns, coordinating the local specification of fascin binding positions and poses across higher-order assemblies.

**Unfavorable cross-bridge accumulation restricts bundle size**

We next probed the mechanisms that limit the size of fascin cross-linked bundles, which generally do not exceed 40 filaments[41,42,51,52]. We reasoned that the geometric diversity we observed in our tomograms would also give rise to a range of cross-bridge conformations that vary in their frequency and corresponding energetic favorability. To survey the cross-bridge conformational landscape in our cryo-ET data, we pursued subtomogram averaging and variability analysis. Our semantic segmentation results facilitated automated extraction of subtomograms well-centered on fascin molecules (Extended Data Fig. 7). Subtomogram averaging produced a 6.7-Å consensus reconstruction of two filaments cross-linked by fascin (Extended Data Fig. 7a,b,d,e and Extended Data Table 3), with 3D classification indicating minimal false-positive picks, validating our denoising approach (Methods). Subsequent multibody refinement revealed flexibility modes highly similar to those we observed

with single-particle analysis. Specifically, hinging motions dominated variability, whereas purely translational displacements were minimal (Extended Data Fig. 7c,f). The deviation of each subtomogram from the consensus structure was measured using the Mahalanobis distance, a metric that allowed us to simultaneously consider all eigenvectors corresponding to rotational displacements (Methods and Extended Data Fig. 7g). We then mapped these scores back to the corresponding particle positions in the full tomograms to generate spatial maps of relative cross-bridge favorability across bundles (Fig. 7a).

Inspection of these maps did not reveal a clear pattern at the individual facsin level, consistent with the extensive geometric heterogeneity of bundles (Extended Data Fig. 6a–c). Because the overall cohesion between filaments in a bundle is mediated by the cumulative action of the cross-linkers at each interfilament interface, we examined variations at this intermediate scale. To simplify the analysis, we analyzed 24 bundle regions of uniform length. The interquartile range for the number of filaments per bundle was 23–43 filaments. We reduced each region to a graph representation consisting of filament nodes and interface edges. We then calculated an interface score for each edge, which is a function of the number of cross-bridges and their average Mahalanobis distance (Methods and Fig. 7a). This revealed variations in interface favorability that were irregularly distributed across bundles.

To investigate how nonuniform interface stability impacts bundle organization, we performed hierarchical clustering (Fig. 7b), a procedure that sequentially divides each bundle graph into smaller clusters through preferential partitioning across unfavorable interfaces. We analyzed the resultant dendrograms through the metrics of average cluster transitivity and modularity, which serve as proxies for cluster stability that are independent of cluster size above a minimum threshold of three filaments (Fig. 7c). Transitivity reflects the number of connected triangles, the basis for hexagonal tiling, whereas modularity reflects the strength of connections within clusters versus between them. When analyzed across bundles, both metrics feature a distinct peak at intermediate linkage distance, suggesting optimum stability of clusters smaller than full bundles. Their average sizes ranged from 4 (transitivity peak) to 8 (modularity peak) filaments, with a minimum of 3 (transitivity peak) and a maximum of 15 (modularity peak; Fig. 7c,d). The boundaries between clusters at the modularity peak corresponded to the worst interfaces (Fig. 7d), suggesting these represent weak points in bundles. These data support a model in which lateral bundle expansion results in the accumulation of unfavorable interfaces, with larger bundles being composed of irregular mosaics of stable clusters. Although this analysis does not provide a quantitative explanation for the distribution of fascin bundle sizes, it does support the existence of an energetic barrier to their indefinite expansion, an intrinsic size restriction mechanism imposed by the variable molecular architecture of bundles.

## Discussion

This study integrates atomistic structural characterization of the flexible fascin–F-actin interface with network-level analysis of bundle architecture to uncover multiscale mechanistic principles for the construction of a supramolecular cytoskeletal assembly (Fig. 7e). We provide detailed insights into the conformational dynamics underlying fascin's regulated F-actin engagement and their allosteric inhibition by G2, a foundation for guiding optimized fascin inhibitor development. At the network scale, fascin's two F-actin binding poses and cross-bridge conformational flexibility facilitate incorporating helical actin filaments into hexagonal bundles by accommodating a broad range of interfilament rotation angles. The cryo-ET denoising procedure we introduced here enabled these insights, allowing us to establish explicit links between protein-level structural plasticity and bundle network geometry. Nevertheless, our denoising approach has limitations. Previous structural knowledge of all system components is required, and it is therefore primarily applicable to reconstituted preparations. Nevertheless, our work highlights the potential for structurally dissecting

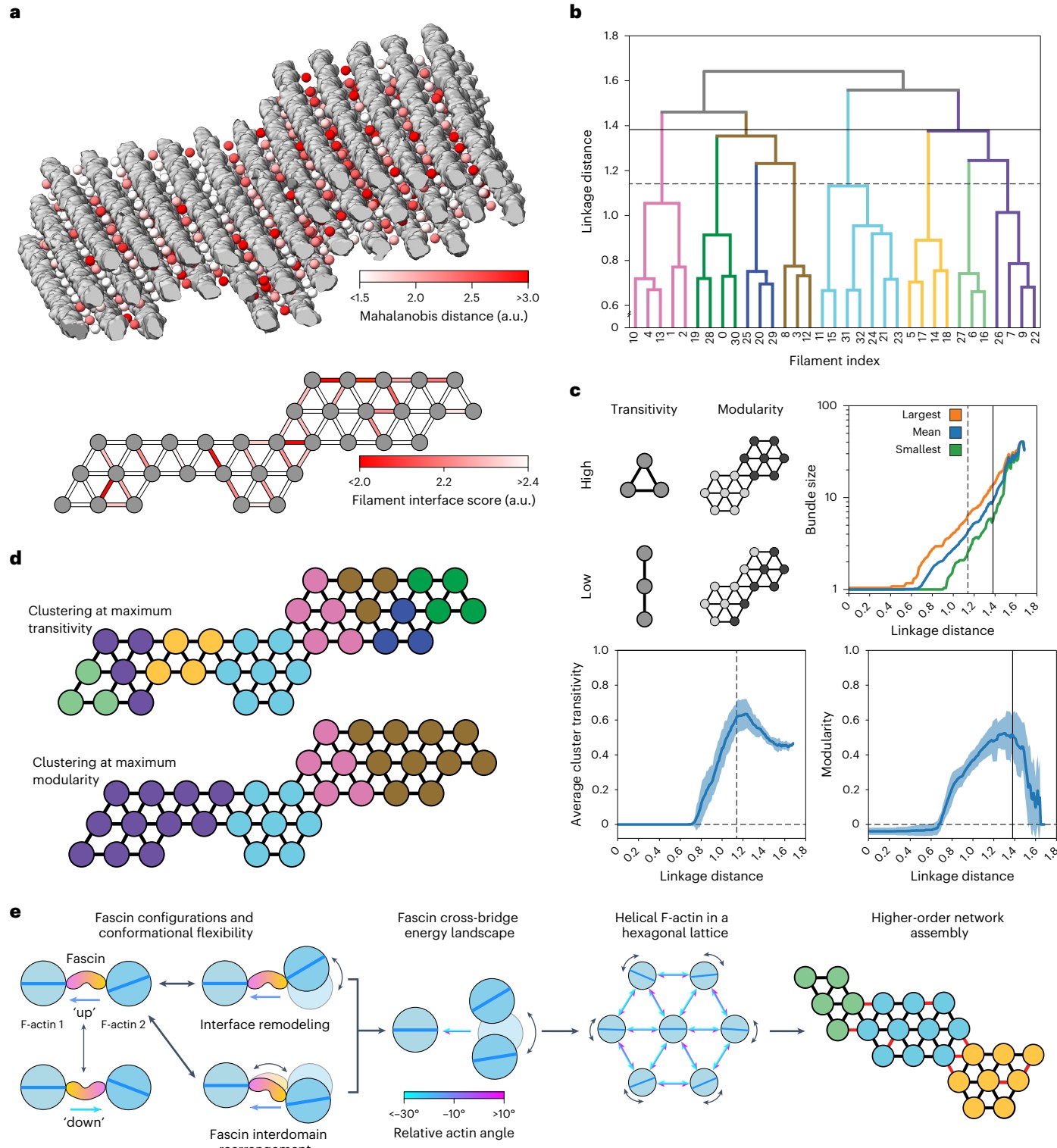

**Fig. 7 | Accumulation of unfavorable cross-bridges restricts bundle growth. a**, Upper, oblique view of fascin cross-bridge Mahalanobis distances calculated from multibody refinement (colored spheres) mapped on to the F-actin bundle. Filaments are displayed in gray. Lower, bundle filament interface scores represented as a weighted graph. **b**, Dendrogram of graph-based hierarchical clustering of the bundle represented in **a**; colors represent clusters at linkage distance corresponding to maximum transitivity. **c**, Plots of graph metrics versus linkage distance. Upper right, quantification of largest, mean and smallest average cluster sizes across bundles. Lower, average cluster transitivity (left) and modularity (right); error bars indicate s.d. Vertical dashed and solid lines indicate linkage distances with maximum transitivity and modularity, respectively, for the bundle shown in **a**. $N = 24$ bundles. **d**, Graphical representation of the clustering at maximum transitivity (upper) and modularity (lower) for the bundle shown in **a**. **e**, Cartoon detailing the link between nanoscale conformational flexibility of the fascin cross-bridge and variable micrometer-scale bundle architecture, which limits bundle size.

macromolecular ensembles composed of hundreds to thousands of proteins, where functional conformational dynamics can emerge that would remain inscrutable in isolated components.

We furthermore delineate how flexible fascin–F-actin network assembly gives rise to long-range binding patterns and restricts bundle size through the introduction of unfavorable interfaces. These phenomena emerge from links between fascin's conformational landscape and variable bundle architecture. Although we focused on ground-state network geometry, filopodia primarily function in mechanically active environments. Forces that deform fascin cross-linked bundles could alter their internal molecular structure and functional properties, for instance by rearranging the F-actin helix through twist-bend coupling[58,61], as well as by modulating fascin's conformational landscape. In addition, whereas the minimal two-component fascin and F-actin system recapitulates much of the previously described complexity of filopodia cores[7,10], fascin collaborates with additional cross-linkers to build other protrusions, such as stereocilia[62,63]. One such cross-linker is plastin, whose conformational dynamics differ substantially from those of fascin[6,64,65], suggesting that multicross-linker networks may feature distinct architectures and functional properties. Indeed, such multicross-linker bundles to grow to larger sizes through unclear mechanisms[41]. We anticipate that the multiscale structural approach we introduce here will enable dissection of multicomponent cytoskeletal networks, allowing detailed examination of how their nanoscale conformational dynamics intersect with their mesoscale architecture, mechanical properties and cellular functions.

## Online content

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

## Methods

### Protein expression and purification

The coding sequence of human fascin-1 (GenBank, NM_003088.4) was obtained from Addgene (cat. no. 31207) and subsequently inserted into a pGEX-6p-1 vector (Cytiva, cat. no. 28954648) using Gibson Assembly (New England Biolabs, cat. no. E2611L) with the following primers: forward, 5′-CTGTTCCAGGGGCCCCTGGGATCCATGACCGCCAACGGCACAGC-3′, reverse, 5′-AGTCACGATGCGGCCGCTCGAGCTAGTACTCCCAGAGCG AGG-3′. The resulting construct was expressed in *Escherichia coli* BL21(DE3) cells (New England Biolabs, cat. no. C2527H) at 18 °C for 16 h after induction with 0.2 mM IPTG (GoldBio, cat. no. I2481C100). The cells were resuspended and disrupted in lysis buffer (25 mM Tris–HCl pH 8.0, 150 mM NaCl). After centrifugation at 48,000*g* for 30 min in a JA-25.50 rotor (Beckman Coulter), glutathione *S*-transferase (GST)-tagged fascin in the supernatant was enriched using GST affinity beads (Glutathione Sepharose 4 Fast Flow, Cytiva). The GST tag was removed by incubating the beads with 3C protease[66] at 4 °C overnight. The cleaved target protein was purified on a HiTrap Q HP ion exchange column (Cytiva) and further polished on a Superdex 200 10/300 Increase column (Cytiva) equilibrated with 25 mM Tris–HCl pH 8.0, 150 mM NaCl and 3 mM DTT.

Chicken skeletal muscle actin was purified as previously described[67] and stored at 4 °C in G-Ca buffer (2 mM Tris–HCl pH 8.0, 0.5 mM DTT, 0.1 mM CaCl$_2$, 0.2 mM ATP, 0.01% NaN$_3$). F-actin was polymerized at room temperature for 1 h by mixing 5 µM monomeric actin in G-Mg buffer (2 mM Tris–HCl pH 8.0, 0.5 mM DTT, 0.2 mM ATP, 0.1 mM MgCl$_2$) with KMEI buffer (50 mM KCl, 1 mM MgCl$_2$, 1 mM EGTA, 10 mM imidazole pH 7.0) supplemented with 0.01% Nonidet P40 substitute (Roche). The polymerized F-actin was stored at 4 °C overnight before grid preparation.

### Cryo-electron microscopy and cryo-electron tomography sample preparation

To prepare fascin cross-linked F-actin bundle for cryo-EM analysis, 0.6 µM F-actin was incubated with fascin at a molar ratio of 1:2 at room temperature for 30 min. For cryo-ET analysis, F-actin bundles were prepared by incubating 0.3 µM each of F-actin and fascin at room temperature for 72 h. Then 4 µl of the specimen was applied to a freshly plasma cleaned (Gatan Solarus, H$_2$/O$_2$ mixture) C-flat 1.2/1.3 holey carbon Au 300 mesh grid (Electron Microscopy Sciences) in a Leica EM GP plunge freezer operating at 25 °C and 100% humidity. After 1 min of incubation, the grid was blotted from the back with a Whatman no. 5 filter paper for 4 s and plunge-frozen in liquid ethane cooled by liquid nitrogen.

### Cryo-electron microscopy and cryo-electron tomography data acquisition

Cryo-EM data were collected on a Titan Krios microscope (Thermo Fisher), operating at 300 kV and equipped with a Gatan K2-summit detector in super-resolution mode using SerialEM[68]. Frame sequences (videos) were recorded at a nominal magnification of ×29,000, corresponding to a calibrated pixel size of 1.03 Å at the specimen level (super-resolution pixel size of 0.515 Å per pixel). Each exposure was fractionated across 40 frames with a total electron dose of 61 e$^-$/Å$^2$ (1.53 e$^-$/Å$^2$ per frame) and a total exposure time of 10 s. The defocus values ranged from −0.8 to −2.2 µm. To alleviate the effects of potential preferential orientation of fascin within F-actin bundles, data were collected with the stage tilted at three different angles: 30°, 15° and 0°. At 30° and 15°, exposures were acquired by targeting a single hole per stage translation. At 0°, exposures were acquired using the beam tilt/image shift strategy, targeting nine holes per stage translation. A total of 6,288 micrographs were collected, with 1,258, 1,439 and 3,591 micrographs obtained at 30°, 15° and 0°, respectively.

Cryo-ET data were collected on a spherical-aberration (C$_s$) corrected Titan Krios microscope operating at 300 kV, equipped with a Gatan K3 detector and BioQuantum energy filter (slit width 20 eV). Tilt series were recorded from −60° to 60° with a tilt increment of 3°.

The image stack at each tilt angle was acquired at a nominal magnification of ×26,000 corresponding to a calibrated pixel size of 2.6 Å at the specimen level (super-resolution pixel size of 1.3 Å per pixel). Each stack was fractionated into 12 frames with a total electron dose of 2.66 e$^-$/Å$^2$ (0.22 e$^-$/Å$^2$ per frame) and a total exposure time of 0.6 s. All 27 tilt series were collected at a nominal defocus of −4 µm.

### Cryo-electron microscopy image processing

Frame sequences were motion corrected, dose weighted and summed with 2 × 2 binning (to a 1.03 Å pixel size) using MotionCor2 (ref. 69). Contrast transfer function (CTF) estimation was performed with CTFFIND4 (ref. 70) using nondose weighted sums. Particle picking was performed using a previously reported neural network-based approach we developed for handling F-actin bundles[65]. Briefly, PDB 7R8V was expanded along its helical axis to produce a filament with 49 protomers and converted to a volume using the molmap command in Chimera[71]. Synthetic data were procedurally generated by spawning bundle pairs from two of copies of this volume featuring a horizontal displacement randomly sampled from a Gaussian distribution centered at 113.8 Å with an s.d. of 15.9 Å (estimated by manually measuring the interfilament distances of 36 bundles in our dataset) and a vertical displacement randomly sampled from a uniform distribution between −180 and +180 Å. A random angular skew sampled from a Gaussian distribution with a mean of 0° and an s.d. of 1.5° was also applied to the second filament.

To generate synthetic images, none, one, two or three of these bundle pairs were loaded while applying a translation in the *x*, *y* and *z* dimensions, each randomly sampled from a uniform distribution between −250 and +250 Å. This composite volume was projected along the *z* dimension to generate a noiseless projection. The first (rot) and third (psi) Euler angles were randomly sampled from a uniform circle in degree increments and the second (tilt) Euler angle was randomly sampled from a Gaussian distribution with a mean of 0° and an s.d. of 7.5°. These projections were corrupted by a theoretical CTF and pink noise using EMAN2 functions[72]. To generate training data for semantic segmentation, the noiseless projections were lowpass-filtered to 40 Å, a binarization threshold of 0.9 was applied and morphological closing was performed by first dilating by 66 Å then eroding by 33 Å. A denoising autoencoder neural network featuring the same architecture as previously described[65] was trained using 150,000 noisy and noiseless projection pairs and a 90:10 training to validation split with a learning rate of 0.00005 until convergence after 14 epochs with a validation cross-correlation coefficient loss of 0.9296. This pretrained network was then trained for semantic segmentation using 50,000 projection pairs and a 90:10 training to validation split with a learning rate of 0.00005 until convergence after 13 epochs with a validation categorical cross-entropy of 0.787. The trained neural network for semantic segmentation was used for inference on the experimental images to identify pixels containing F-actin bundles. Tiles of 192 pixels spaced in 48-pixel increments from micrographs binned by 4 were passed through the neural network for semantic segmentation before being stitched back together via a maximum intensity operation. From these semantic maps, particle picks were generated by removing overlap within a 60 Å distance.

The coordinates of 3,056,360 picked particles were imported into RELION v.4.0 (ref. 73) and extracted with a box size of 448 pixels, then binned by 2. The extracted particles were imported into cryoSPARC v.4.2 (ref. 74) for reference-free two-dimensional classification. A subset of 2,231,774 particles were selected from classes featuring parallel actin filaments. A quarter of them (550,000) were used for ab initio 3D reconstruction to generate an initial reference. This reconstruction featured seven filaments arranged in a hexagonal lattice with several apparent interfilament densities corresponding to fascin cross-bridges (Extended Data Fig. 1a). The full selected set of 2,231,774 particles were then subjected to masked homogeneous refinement using a cylinder-shaped mask covering all 7 filaments. The aligned

particles were subsequently reimported into RELION for 3D classification without image alignment. Particles from the class displaying the clearest hexagonal arrangement were selected, recentered on the fascin cross-bridge with the strongest density, re-extracted and reconstructed. An 'H'-shaped mask covering the recentered fascin cross-bridge and its two associated filaments was created. The recentered particles were then subjected to masked 3D classification with a global search range for rot angles and a 10° local search range for psi and tilt angles. Particles with clear cross-bridge density were selected and subjected to another round of focused 3D classification with a mask covering the fascin cross-bridge and its two interacting actin subunits on each filament. Classes with improved fascin density were pooled and refined using the 'H'-shaped mask, yielding a density map displaying strong fascin density in the mask and four neighboring weak fascin densities.

Particles were subjected to symmetry expansion with recentering on each of the additional fascin densities, followed by focused 3D classification. Particles corresponding to classes with strong cross-bridge density were then re-extracted without binning at a pixel size of 1.03 Å. After duplicate removal, CTF refinement, Bayesian polishing, 3D refinement and postprocessing, we obtained a consensus 3.4-Å resolution density map featuring streaking artifacts in fascin (Extended Data Fig. 1). To further improve the map quality, two rounds of multibody refinement were performed, alternately masking fascin and one filament as one body and the other filament as the other body (that is, F-actin 1 + fascin as body 1 and F-actin 2 as body 2; then F-actin 2 + fascin as body 1 and F-actin 1 as body 2). This greatly improved the density of each F-actin–fascin interface when it was in the body 1 mask. The final 3D refinements yielded postprocessed density maps with resolutions of 3.1 and 3.0 Å, respectively.

To generate a composite map containing both well-resolved fascin–F-actin interfaces, the two multibody maps were aligned on fascin. The poorly resolved pair of β-trefoil domains from each map was removed with the 'split map' command, and the remaining well-resolved density from the two maps was stitched together using the 'vop maximum' command, both in UCSF Chimera[71].

To resolve a fascin cross-linked F-actin hexagonal bundle element, the original 3,056,360 picked particles were extracted in RELION using a box size of 448 pixels and binned by 8. Particles were imported into CryoSPARC and aligned by homogeneous refinement with a cylinder-shaped mask and a hexagonal bundle reference generated from the abovementioned ab initio 3D reconstruction. Aligned particles were subjected to several rounds of focused 3D classification in RELION to enrich particles with strong densities for all possible fascin molecules and F-actin filaments in a hexagonal bundle element. The 8,477 particles contributing to the class featuring the best density were re-extracted with a box size of 448 pixels and no binning, as well as a box size of 720 pixels binned by 2. Both particle stacks were reconstructed and refined, generating postprocessed density maps at 8.7 and 12.0 Å resolution, respectively. Four additional classes were refined that featured high-quality density for the central filament and discernable density for its six bound fascins, but varied quality for the peripheral filaments and cross-bridges.

Local resolution estimation was performed using the procedure implemented in RELION.

### Measuring hexagonal bundle element filament parameters

Densities corresponding to the seven actin filaments in the subnanometer resolution 460-Å bundle reconstruction were segmented using Chimera. The rotational phase offsets between each side filament and the central filament were measured by individually fitting the density of each side filament into the central filament using Chimera's 'Fit in Map' tool. The consensus rise and twist of each filament was measured using both the relion_helix_toolbox[75] in RELION v.4.0 and the iterative helical real space reconstruction program hsearch_lorentz[76]. To

prepare the filaments for analysis with these programs, the segmented filament volumes were resampled such that the filament was centered and aligned along the proper axis. The programs were seeded with initial rise and twist estimates of 27.9 Å and −166.0°, respectively. For local twist measurements, atomic models of actin subunits (PDB 7R8V) were individually rigid-body fit into each filament density to generate a docking model, which was passed as input to an updated version of a custom script that we recently reported[58].

Equivalent docking models generated from the centered and aligned volumes were furthermore used to perform axial shift measurements. The $C_\alpha$ of H plug residue I266 was selected as a fiducial, and the $x, y, z$ coordinates of this atom in the central subunit of each model were recorded. Axial shift was calculated as the difference in this atom's $z$ coordinate between each of the side filaments and the central filament, equivalent to displacement along the filament axis.

### Tilt series reconstruction

Motion correction of individual frames and CTF estimation of combined tilt stacks were performed in WARP[77]. Data were exported for subsequent alignment and reconstruction using the IMOD software package[78]. Tilt series were binned by 3 to a final pixel size of 7.8 Å, aligned with the patch-tracking approach, then reconstructed using weighted back-projection with a SIRT-like filter equivalent to 12 iterations.

### Synthetic tomography data generation

Synthetic datasets approximating subtomograms were generated to train neural networks for denoising and semantic segmentation. The training datasets for the neural networks consisted of noisy source volumes simulating empirical data, along with paired target volumes of in silico-generated ground truth fascin–F-actin bundles. To produce these volumes, the seven-filament bundle element reconstruction was laterally expanded to produce a larger hypothetical lattice of 19 filaments bound by 84 fascins. Atomic models of F-actin (PDB 7R8V) and fascin cross-bridges were fit into this extended volume, and converted to volume data using the molmap command in Chimera. These docked components were used to generate plausible regions of bundles with varying architectures. For each simulated region, a random subset of the F-actin volumes was loaded. Fascins were loaded only if they were in a position to bind these filaments; to simulate sub-stoichiometric binding, a random subset of these fascins were rejected. All of the F-actin volumes were combined in a single volume, all of the fascin volumes were combined in a separate volume and both volumes were saved. A library of 200 volume pairs was procedurally generated using this approach.

The library was used to generate a dataset of 128 voxel training volumes at a sampling of 7.8 Å per voxel. Paired fascin and F-actin volumes were rotated about the rot and psi angles by random, uniformly sampled values between 0° and 359°, whereas the tilt angle was randomly sampled from a Gaussian probability distribution centered at 90° with an s.d. of 20°. The density was translated in the box along each dimension by a random uniform translation in the range of ±195 Å.

A set of 20,000 synthetic noisy volumes were generated by first projecting the repositioned volume at fixed angles corresponding to the experimental tilt series collection scheme (−60° to +60° with a 3° increment). These projections were corrupted by the CTF, and reconstructed back into a 3D volume using reconstructor class functions as implemented in the EMAN2 python package[72]. Empirical noise was extracted from each of our experimental tomograms by computing the average Fourier transform of 50 randomly sampled 128 voxel boxes. For each synthetic volume, one of these empirical noise boxes was randomly selected, multiplied element-wise by a white noise box of the same dimensions, normalized and scaled by a random scale factor that modulated the signal-to-noise ratio of each synthetic particle. The synthetic volume was then summed with its noise volume in Fourier

space. To account for interpolation artifacts from CTF application or noise addition in Fourier space, the synthetic volumes were cropped to 64 voxels in real space. Each noisy particle had a corresponding noiseless ground truth particle, as well as binarized semantic maps calculated from the paired F-actin and fascin volumes.

## Neural network training for tomogram denoising and semantic segmentation

Pretraining of a denoising autoencoder consisting of 3D convolutional layers in a U-net architecture (Extended Data Fig. 5) was performed with a single NVIDIA A100 graphics processing unit (GPU) with 80 GB of video RAM, using a learning rate of 0.0001. Training was run on the 20,000 pairs of noisy and ground truth volumes with a 90:10 training to validation split, until the network converged with a cross-correlation coefficient validation loss of 0.8659 after 20 epochs. Initial inference testing on experimental tomograms resulted in distorted denoised volumes. We reasoned that this was because the synthetic data did not model corruption of the experimental data with sufficient fidelity. We therefore continued to train the network using a domain adversarial neural network approach[79]. An additional network head was added for domain classification by forking the network output after the feature extraction layers (Extended Data Fig. 5a). The domain classification head consists of a gradient reversal layer and additional 3D convolutional layers, which are followed by flattening to a dense layer, then a binary classification layer with a sigmoid activation function. One hundred synthetic volumes and 100 volumes extracted from the experimental tomograms were used as the training set for the domain classifier. Adversarial training was performed by alternatively passing these data through the domain classifier head, followed by retraining the feature extractors with the denoising head using only the 100 synthetic volumes. This adversarial training was run for ten iterations; however, after multiple iterations, too many high-resolution details were lost, so the first iteration of adversarial training was used for denoising.

After using a domain adversarial neural network to complete training of the denoising autoencoder, a semantic segmentation network was trained using 10,000 volume sets and the autoencoder's pretrained weights. The final layer was adapted to produce multichannel outputs, featuring a softmax activation layer and random initial weights. After training for ten iterations with a learning rate of 0.00001, the categorical cross-entropy loss of the semantic segmentation network was 0.0556.

## Tomogram denoising and semantic segmentation

The trained neural networks were used to denoise and semantically segment empirical tomograms, once again using single A100 GPUs. The tomograms were extracted into 64-voxel tiles sampled every 32 voxels in $x$ and $y$ and every 4 voxels in $z$, normalized and passed as inputs to the neural network. The network outputs were masked with a 48-voxel cubic mask to minimize edge artifacts and stitched together via maximum intensity projection. Each input tomogram produced three outputs: a denoised tomogram, a semantic map of actin filaments and a semantic map of fascin molecules.

## Measurement of F-actin rotational phase offsets in denoised tomograms

The denoised tomograms were manually inspected to identify bundles with high-quality density, straight filaments and uniform fascin cross-bands spanning multiple crossovers. Subsequently, a 17-subunit F-actin model was generated from PDB 7R8V and fit into filament densities in the bundle using Chimera while maintaining identical axial register of the filaments across the bundle. Synthetic density maps were then generated for each fitted F-actin atomic model using the 'molmap' command in Chimera with a resolution cutoff of 6 Å. The rotational phase offset was measured by fitting the synthetic density map of the rotating filament to the reference filament using Chimera.

## Subtomogram averaging

Subtomogram particle picking required minimal postprocessing of the fascin semantic maps. A threshold of 0.9 was applied to each semantic map, and the centroids of objects larger than 50 voxels were designated as potential fascin picks. The center of the hole in the carbon film for each tomogram was manually picked, and potential fascin picks within a 740-voxel radius of the hole center were retained as picks. Subtomogram averaging was performed using RELION v.4.0. From 26 tomograms, 135,000 192-voxel pseudo-subtomograms were extracted at bin 3 (voxel size 7.8 Å) and cropped to a 64-voxel box. A small subset of 1,000 pseudo-subtomograms was used for initial model generation with three classes. Two of the three classes contained 82% of the particles, and the most populous class was selected. An initial global alignment of particles was performed using Class3D with one class. A subset of 5,000 particles with tilt angles outside the range 60–120° were subsequently excluded. The rot angle was then removed from the metadata, and a bimodal psi prior was used for subsequent 3D auto-refinement. An H-shaped mask (85% $z$-length) was used for focused 3D auto-refinement and subsequent processing of the central filaments and fascin bridge. Three-dimensional classification using two classes was attempted to remove partially decorated fascin monomers or picks of single filaments, but both classes were essentially identical, suggesting a minimal false-positive rate in picking.

After another round of local 3D auto-refinement, pseudo-subtomograms were sequentially re-extracted and subjected to local 3D auto-refinement at bin 2, then bin 1. This produced a reconstruction with a nominal resolution of 8.5 Å, yet the map appeared distorted. After two rounds of CTF refinement, FrameAlignTomo and local 3D auto-refinement, a final undistorted 6.7-Å reconstruction was obtained. Multibody refinement was conducted in an identical manner as described above for the single-particle analysis. Local resolution estimation was performed using RELION's implementation.

## Bundle hierarchical clustering based on subtomogram variability analysis

Per-subtomogram deviations from the consensus reconstruction were measured using the Mahalanobis distance of each particle's multibody refinement amplitude along the eigenvectors corresponding to rotations. To calculate filament interface scores, fascin cross-bridges were assigned to interfilament interfaces using a custom script. The filament interface score (FIS) was computed as:

$$\mathrm{FIS} = (N/L)/((1 + \bar{E}))$$

where $N$ is the number of fascin cross-bridges along the interface, $L$ is the length of the interface and $\bar{E}$ is the average energy (Mahalanobis distance) of the cross-bridges along the interface. Graphs were constructed from 24 well-ordered bundle regions approximately 400 nm in length, where actin filaments correspond to nodes, and nodes are joined by an edge if the corresponding actin filaments were bridged by at least one fascin in the region. Edge weights were assigned as the filament interface scores.

Hierarchical clustering was performed using the ward linkage method, and the maximum distance was normalized across all graphs to enable direct comparisons. The metrics of transitivity, modularity and minimum, average and maximum cluster sizes were computed as a function of linkage distance for each graph.

## Atomic model building, refinement and analysis

Previously determined structures of actin (PDB 7R8V)[67] and fascin (PDB 3LLP)[27] were rigid-body docked into each of the high-resolution multibody refinement-derived density maps in ChimeraX v.1.6.1 (ref. 80), then flexibly fitted with ISOLDE[81]. Atomic models were refined using phenix.real_space_refine[82] alternating with manual adjustments in Coot[83]. Refined models were rigid-body fit into the composite map,

split at the boundary between the well-resolved and poorly resolved β-trefoil domains in the multibody map from which the model was derived, merged and real-spaced refined. Model validation was conducted with MolProbity[84] as implemented in Phenix. The structural rearrangements of fascin in different states were analyzed with the DynDom online server[85].

### Computational model

Based on the cryo-EM multibody refinement analysis (Fig. 5a), a computational model was developed to predict the 3D arrangement of F-actin arrays cross-linked by fascin. The code was implemented in MATLAB. This model iteratively adjusts the angular shift of each F-actin in the array to find the configuration corresponding to the maximum probability of forming fascin cross-bridges (Extended Data Fig. 4a), which is inferred from an energy function linked to cross-bridge geometry (Extended Data Fig. 4b). The angular shift between a pair of filaments determines the physically plausible fascin bridging positions. The favorability of a bridging position is furthermore linked to the extent a fascin cross-bridge must be deformed to accommodate the local binding geometry, as visualized in the multibody analysis. Both of these physical phenomena are simultaneously represented in the energy function, where distances between fiducial actin residues spanning the fascin–actin contact surface serve as a proxy for the local interface geometry (Extended Data Fig. 4b).

### Derivation of geometric rules

Four pairwise distances between fiducial residues in fascin–actin contact sites were used to estimate cross-bridging energies: $R95_i$ and $R95_{j+1}$, $R95_i$ and $S350_{j+3}$, $D24_{i+2}$ and $S350_{j+3}$, and $D24_{i+2}$ and $R95_{j+1}$ (Extended Data Fig. 4b). The model assumes that the distances between the fiducial residues are independent. In addition, the distance between the center of geometry of these residues and the plane crossing the filament axes was incorporated to prevent unrealistic fascin distortions.

The distance-dependent cross-linking probability $P_n(d)$ for each distance $n$ (where $n = 1, 2, …, 5$), was calculated as follows:

$$P_n(d) = e^{-\frac{1}{2}\left(\frac{d-\mu}{m\sigma}\right)^2}$$

Here, $\mu$ and $\sigma$ represent the mean and s.d. of $d$, respectively. These parameters are calculated from multibody refinement analysis and are reported in Extended Data Table 2. Parameter $m$ is the multiplication factor for the s.d. $\sigma$, which encodes cross-bridge plasticity. The total cross-bridging probability $P$ is given by the product of the five $P_n$:

$$P = P_1 \times P_2 \times P_3 \times P_4 \times P_5$$

Distributions of these probabilities as a function of $d$ are presented in Extended Data Fig. 4c,d. The normalized cross-bridge energy was calculated as:

$$E_c = -\frac{1}{P}$$

### Algorithm of growing bundles

Each filament in the model is identified by the angle that the first monomer forms with the origin of the domain. Filaments are represented as double-stranded helical structures composed of 14–19 monomers, with each monomer rotated by −166.7° and translated by 2.75 nm along the vertical axis.

The algorithm begins by placing the first filament (filament $A$) at a random position in the bundle and assigning it a random absolute rotation angle $\theta_A$. In the first iteration, a second filament (filament $B$) is positioned 12.15 nm from filament $A$. The algorithm evaluates the probability $P$ of forming a cross-bridge, $P$, for each possible rotation angle $\theta_B$ of filament $B$, which ranges from 0 to 359.9° in increments of 0.1°.

For each value of $\theta_B$ the probability $P$ of forming a cross-bridge between filament $A$ and filament $B$ is calculated. This involves evaluating the probability for all pairs of protomers, specifically ($i$ and $i + 2$) from filament $A$ and ($j + 1$ and $j + 3$) from filament $B$. The probability $P$ is defined as the product of five distance-dependent probabilities $P_1, P_2, P_3, P_4$ and $P_5$, which are based on the distances between specific fiducial residues.

The algorithm then selects the rotation angle $\theta_B$ that maximizes the cross-bridge formation probability $P$, considering only angles where $P$ exceeds a minimum threshold $\tau$ (the minimum threshold for fascin binding). Once filament $B$ is added, the algorithm proceeds to add additional filaments. Each new filament is positioned within 12.15 nm of at least two existing filaments in the bundle. This process continues iteratively, with the algorithm optimizing the rotation angles and positions of new filaments to maximize the likelihood of cross-bridge formation by fascin.

### Model parameterization based on sensitivity analysis

Model parameterization involved adjusting two free parameters, $m$ and $\tau$, based on their effects on the distribution of $P$ (the overall probability of forming a cross-bridge) and on the fraction of filament pairs that formed a fascin cross-bridge. For this analysis, filament pairs with randomized angular shifts were used. First, we evaluated the effect of $m$ on the cross-bridge probability distribution. We tested how systematic variations in $m$ affected the distribution of $P$. For each $m$ value between 1 and 7 (with a step size of 1), and at each value of $\tau$ from 0.1 to 0.8 (with a step size of 0.1), 50 independent runs were performed, resulting in a total of 2,800 runs. The normalized cross-bridge energy distribution exhibited a biphasic relation with $m$: it increased as $m$ increased from 1 to 4, and then decreased for $m > 4$ (Extended Data Fig. 4e). Next, we examined how systematically varying $m$ affected the fraction of cross-bridges formed across runs. We observed a low fraction of cross-bridges for $m < 4$, with an average cross-bridge fraction <0.5 and a median fraction of 0 (Extended Data Fig. 4f). Based on these data, we identified $m = 4$ as the value that captured the variability in cross-linking probabilities and resulted in a high enough fraction of fascin bonds (Extended Data Fig. 4e,f).

The sensitivity of the model outputs was also assessed for $\tau$. For these simulations, $m$ was fixed at 4 as $\tau$ was varied between 0.1 and 0.8 (with a step size of 0.1). For each $\tau$ step, 50 model runs with randomized angular shifts were performed, totaling 400 independent runs. We found that for $\tau > 0.6$, $E_c = 0$, indicating that $\tau$ was too restrictive to allow cross-bridges to form (Extended Data Fig. 4g). For $\tau \le 0.6$, the distribution of $E_c$ broadened, providing a range ($0.1 \le \tau \le 0.6$) that produced fascin cross-bridging (Extended Data Fig. 4g). Further analysis of the mean and median cross-bridging fraction revealed that using $\tau < 0.4$ was too permissive, allowing constitutive bond formation (Extended Data Fig. 4h). Therefore, the acceptable range for $\tau$ was identified as $0.4 \le \tau \le 0.6$. Based on this range, $\tau = 0.4$ was selected for further simulations of higher-order fascin cross-linked assemblies.

### Evaluating the effect of F-actin angular shift on cross-bridge energy

We analyzed pairs of filaments composed of 14 protomers (spanning a single crossover) to evaluate the effect of relative angular shifts on the probability of fascin cross-bridging, indicated as normalized energy, $E_c$. Systematic variation of the two filaments' rotation angles between 0° and 180° identify regions without cross-bridging, corresponding to $E_c = 0$, and regions of cross-bridging with values of normalized $E_c < 0$ (Extended Data Fig. 4i). Consistent with the periodicity of F-actin, we observed a regular pattern of teardrop-shaped energy wells around minima (indicated by white dots in Extended Data Fig. 4j). To explore the implications of this pattern, we examined $E_c$ with the absolute rotation angle of filament $A$ held constant at 102.7° while rotating filament $B$ between 60° and 140°. This revealed one global minimum

in $E_c$ and several local minima (Extended Data Fig. 4j). In this case, the global minimum corresponded to a fascin bridge in the 'down' pose, whereas the most favorable local minimum corresponded to fascin bridge in the 'up' pose at a nearly equivalent position (Extended Data Fig. 4k). This suggests that the two teardrops that intersect at a given filament $A$ angle correspond to nearly isoenergetic 'up' and 'down' binding positions, with the relative favorability modulated by subtle changes to the interfilament rotation angle. To test the hypothesis of isoenergetic binding positions, we examined all minima in angular shift ranges that satisfy $\tau$ to form fascin bonds (yellow in Extended Data Fig. 4l). Consistently, when the rotation angle of filament $A$ is greater than that of filament $B$ (upper diagonal in Extended Data Fig. 4i), the $E_c$ minima correspond to 'down' pose bridges (Extended Data Fig. 4m), whereas when the rotation angle of filament $B$ is greater than that of filament $A$ (lower diagonal in Extended Data Fig. 4i), the $E_c$ minima correspond to 'up' pose bridges (Extended Data Fig. 4n), with adjacent minima corresponding to bridging of adjacent axially shifted protomers.

## Graphics and additional analysis

Figures and videos were generated with UCSF Chimera and ChimeraX. All statistical analysis and plotting were performed in GraphPad Prism. Custom code for tomogram denoising and downstream analysis was generated with the assistance of ChatGPT v.4.0.

## Reporting summary

Further information on research design is available in the Nature Portfolio Reporting Summary linked to this article.

## Data availability

The cryo-EM density maps and atomic models generated in this study have been deposited in the Electron Microscopy Data Bank (EMDB) and PDB: fascin bound filament 1 (EMDB EMD-43364, PDB 8VO5); fascin bound filament 2 (EMDB EMD-43365, PDB 8VO6); composite map of the fascin cross-bridge (EMDB EMD-43366, PDB 8VO7); multibody binned reconstructions eigen_left (EMDB EMD-43367, PDB 8VO8); eigen_middle (EMDB EMD-43368, PDB 8VO9); eigen_right (EMDB EMD-43369, PDB 8VOA); fascin cross-linked hexagonal bundle element with a box size of 460 Å (EMDB EMD-43370); fascin cross-linked hexagonal bundle element with a box size of 740 Å (EMDB EMD-43371); subtomogram average of fascin cross-bridge (EMDB EMD-43372). The trained neural networks for denoising and semantically segmenting micrographs and tomograms as well as the atomic models and volume maps used to generate synthetic datasets are available via Zenodo at https://doi.org/10.5281/zenodo.10456803 (ref. 86). All other data are presented in the manuscript. Source data are provided with this paper.

## Code availability

Custom code is available at https://www.github.com/alushinlab/fascin as open source.

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

## Acknowledgements

We thank J. Sotiris, H. Ng and M. Ebrahim at the Rockefeller University Evelyn Gruss Lipper Cryo-EM Resource Center for their assistance with data collection, and A. Pan (RU) for performing F-actin purification and assistance with preparing an interface for neural network-based particle picking. We thank L. Chin for providing the cDNA of human fascin-1 via Addgene (plasmid 31207). K.H. was supported by a Rockefeller University Anderson Center for Cancer Research postdoctoral fellowship. This work was funded by grants from the National Institutes of Health (NIH; grant no. R01GM141044) and the Alfred P. Sloan Foundation (grant no. G-2020-14047) to G.M.A., as well as NIH grant no. R35GM147491 to T.C.B. This research was also supported by the Stavros Niarchos Foundation (SNF) as part of its grant to the SNF Institute for Global Infectious Disease Research at the Rockefeller University.

## Author contributions

R.G., M.J.R., T.C.B. and G.M.A. designed the research. R.G. purified fascin and collected cryo-EM and cryo-ET datasets. M.J.R. and

R.G. processed cryo-EM data. R.G. built atomic models. R.G. and K.H. reconstructed the tomograms. M.J.R. developed algorithms to denoise and segment tomograms, performed subtomogram averaging, and carried out hierarchical clustering and subtomogram variability analysis. K.R.C. and T.C.B. performed computational simulation studies. R.G., M.J.R., T.C.B. and G.M.A. wrote the paper with input from all other authors.

## Competing interests

The authors have no competing interests to declare.

## Additional information

**Extended data** is available for this paper at https://doi.org/10.1038/s41594-024-01477-2.

**Correspondence and requests for materials** should be addressed to Rui Gong or Gregory M. Alushin.

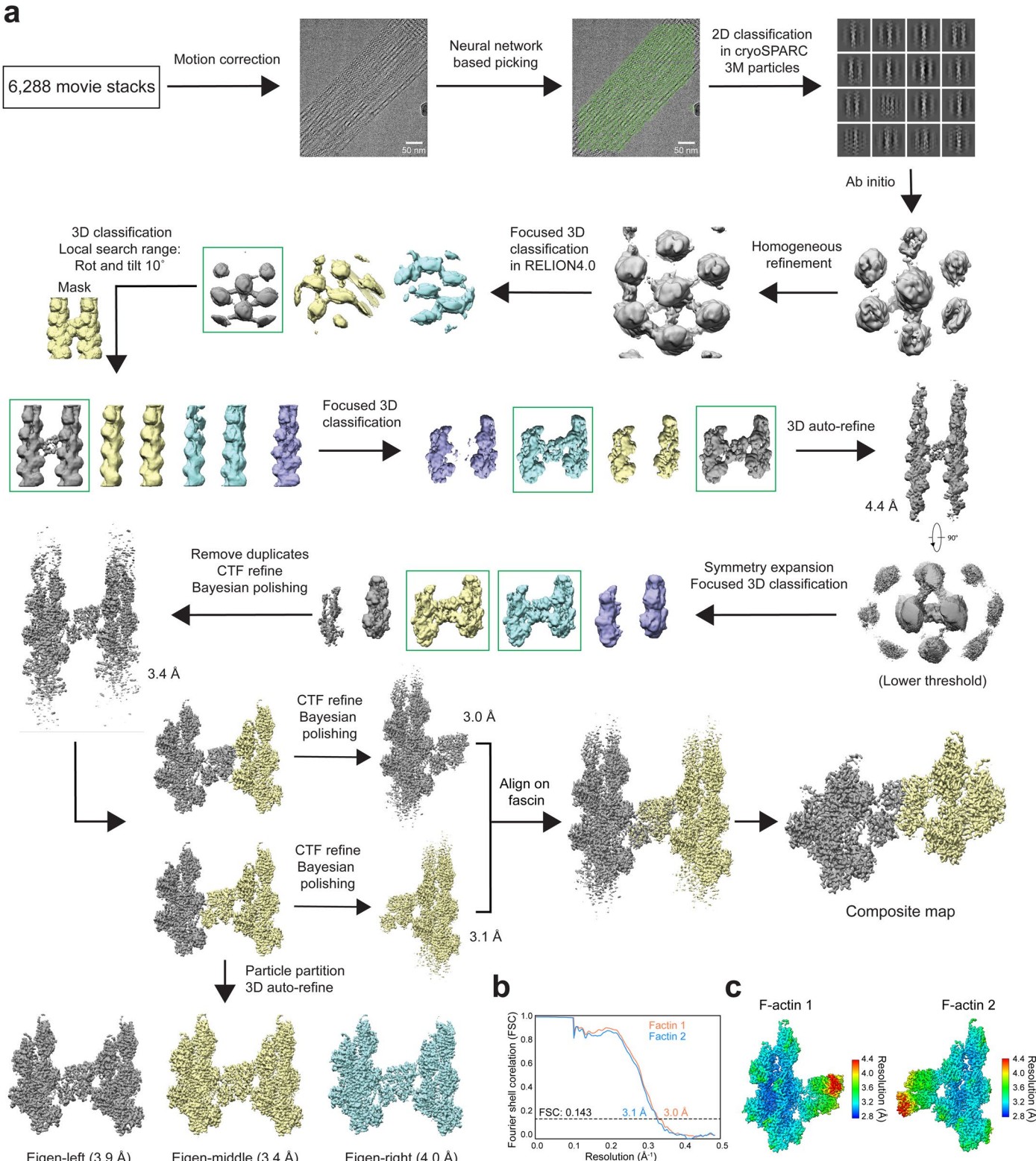

**Extended Data Fig. 1 | Cryo-EM data processing workflow. a**, Scheme of high resolution fascin crossbridge reconstruction data processing, including multi-body refinement. **b**, Fourier shell correlation (FSC) curves. **c**, Local resolution assessment.

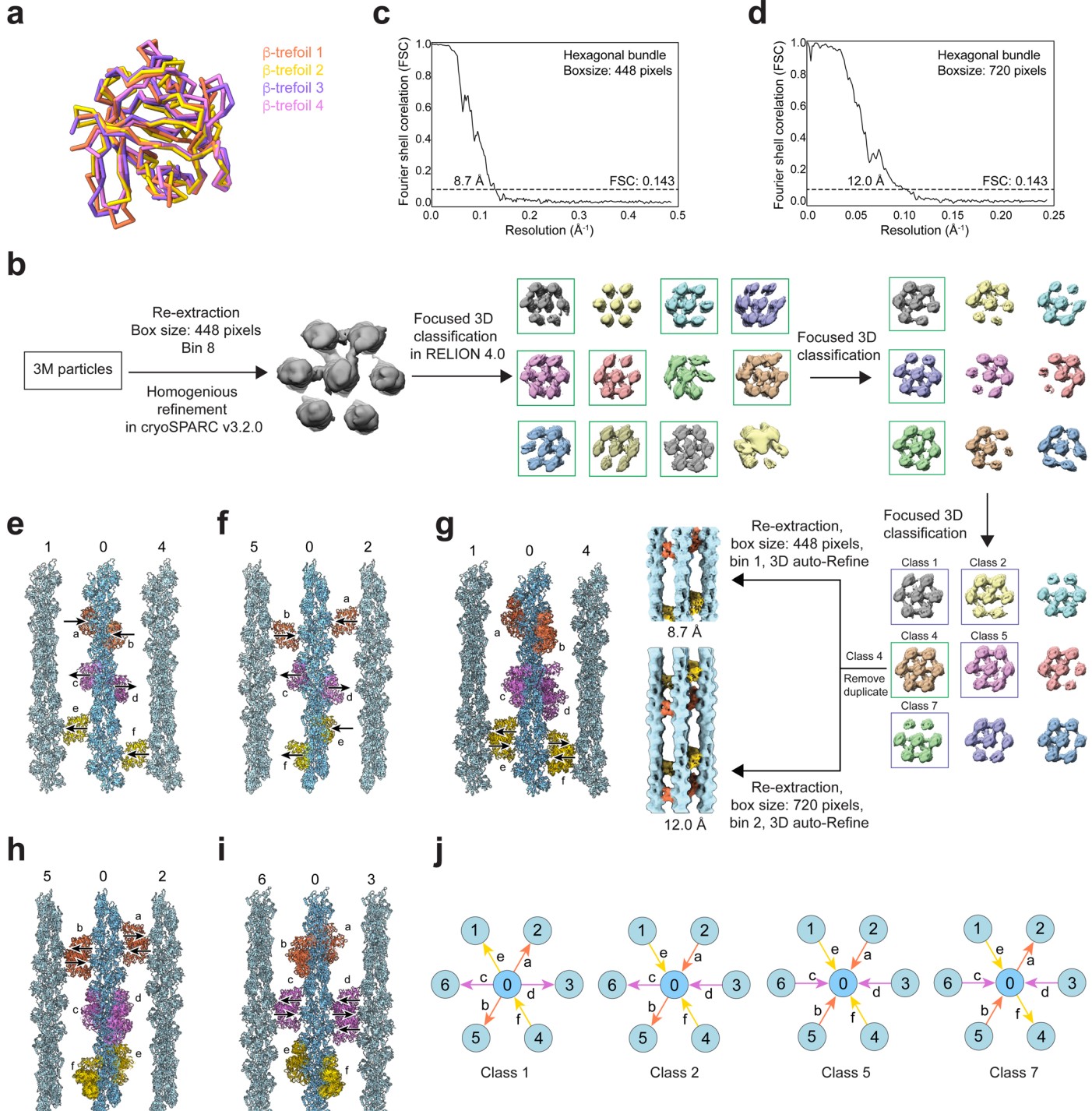

**Extended Data Fig. 2 | Analyses of fascin conformation and bundle element architecture. a**, Superposition of fascin's four individual β-trefoil domains extracted from the consensus atomic model (Fig. 1f). **b**, Cryo-EM data processing workflow for reconstructing a hexagonal bundle element. **c** and **d**, FSC curves for the bundle element reconstructed with two different box sizes. **e** and **f**, Side views of actin planes 1-0-4 (**e**) and 5-0-2 (**f**) from the 8.7-Å reconstruction docking model. **g-i**, Side view of each actin plane of models from 5 bundle element classes aligned on the central filament. **j**, Distributions of poses of the 6 central-filament associated fascins from the four additional bundle element classes.

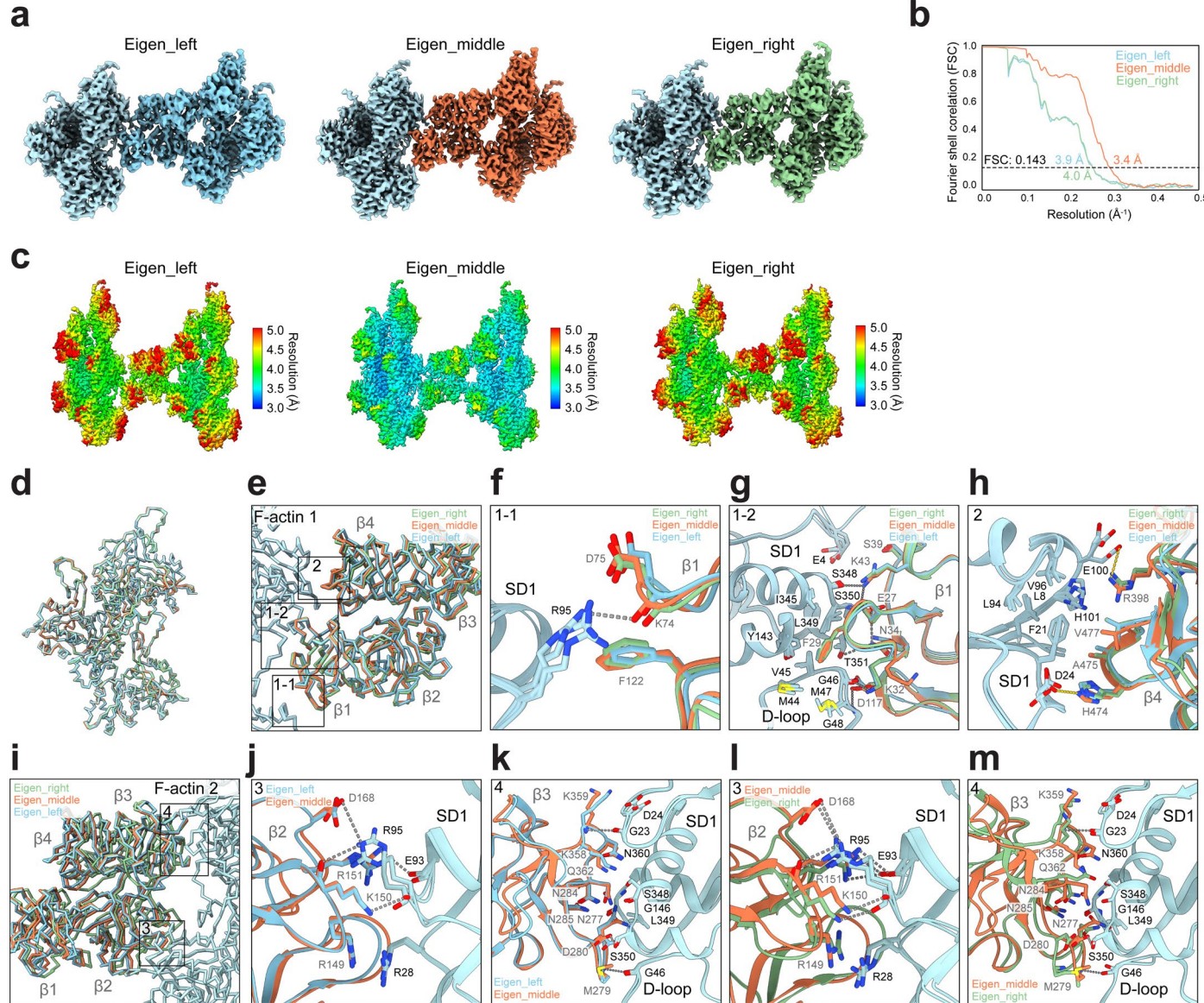

**Extended Data Fig. 3 | Analyses of multi-body derived reconstructions.**
**a**, Cryo-EM density maps of eigen_left, eigen_middle and eigen_right reconstructions. **b** and **c**, FSC curves (**b**) and local resolution assessment (**c**) of the three reconstructions. **d**, Superposition of all six F-actin models from the three reconstructions. All F-actin 1 models are colored light blue, while F-actin 2 models are colored as in Fig. 5b. **e**, Comparison of the fascin-F-actin 1 interface across the three snapshots, superimposed on F-actin 1. **f-h**, Detail views of contacts indicated in **e**. **i**, Comparison of the fascin-F-actin 2 interface across the three snapshots, superimposed on F-actin 2. **j-m**, Detail views of contacts indicated in **i**.

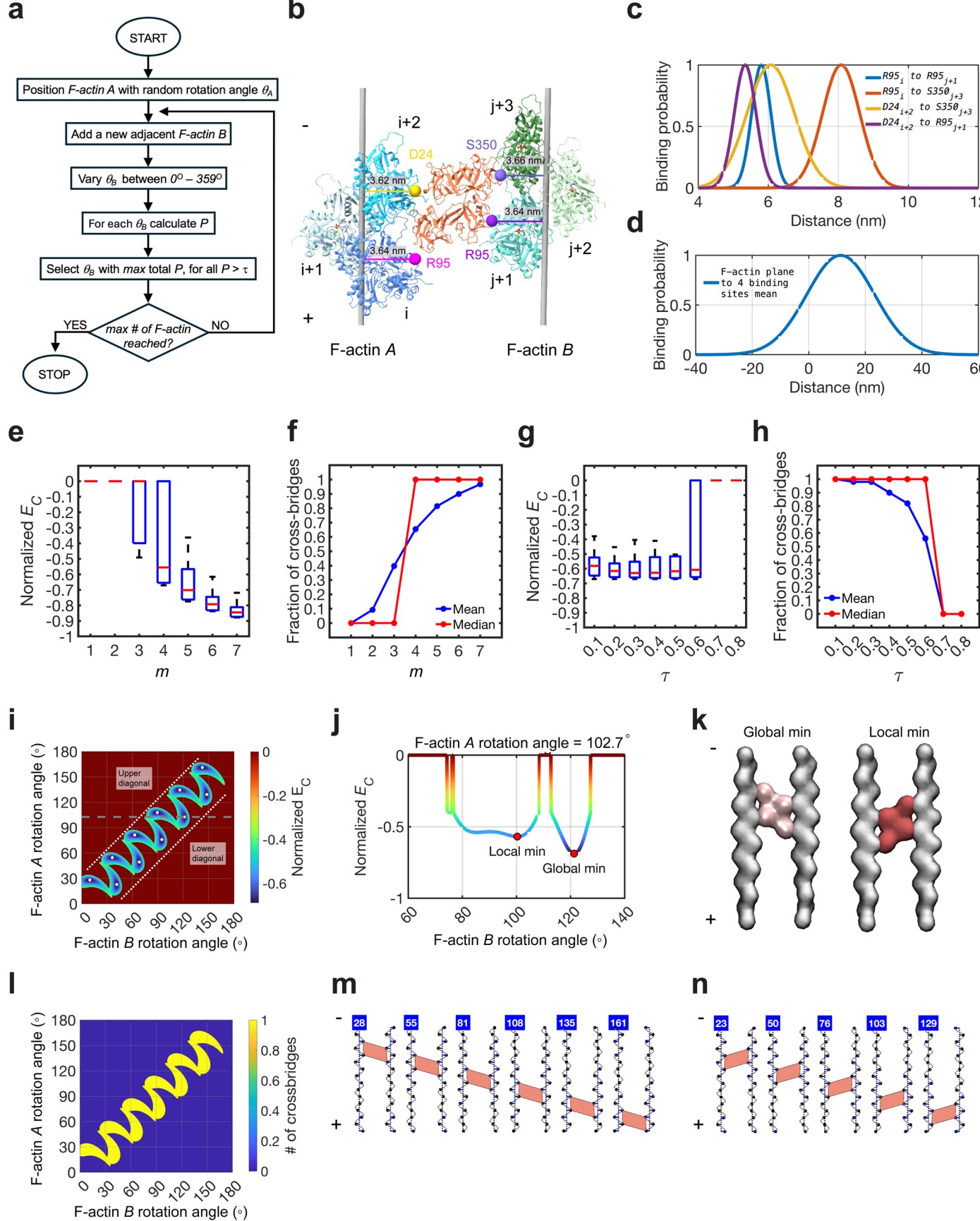

**Extended Data Fig. 4 | See next page for caption.**

**Extended Data Fig. 4 | Computational model parametrization and analyses.**
**a**, Flowchart of the computational model. **b**, Atomic model of the fascin crossbridge; fiducial residues for distance calculations are indicated. Grey cylinders represent F-actin axes. **c**, Fascin binding probabilities as a function of inter-fiducial distances. **d**, Fascin binding probability as a function of the separation between the geometric center of the fiducials and the plane which spans both filament axes. **e**, Distribution of normalized cross-bridge energies versus the standard deviation multiplier, $m$, between 1 and 7. N = 2,800 independent runs. Filament $A$ was initialized with random absolute rotation angles, while $\tau$ was varied systematically between 0.1 and 0.8 with 0.1 increments. Red line indicates median; lower and upper bounds of box are 25th and 75th percentile, respectively. Whiskers indicate range. **f**, Fraction of cross-bridges from simulation results in **e**. **g**, Boxplot of the distribution of normalized cross-bridge energies while varying $\tau$ using $m = 4$. N = 400 independent runs. F-actin $A$ was initialized with random absolute rotation angles. Red line indicates

median; lower and upper bounds of box are 25th and 75th percentile, respectively. Whiskers indicate range. **h**, Fraction of cross-bridges from simulation results in **g**. **i**, Heatmap of the normalized cross-bridge energies for a filament pair as a function of absolute rotation angles. White dots show energy minima. **j**, Plot of normalized cross-bridge energy for F-actin $A$ rotation angle = 102.7° and varying the rotation angle of F-actin $B$ between 60° and 140°, with increments of 0.1°, corresponding to gray dashed line in **i**. **k**, Fascin-cross-linked filament pair for the global and local minimum indicated in **j**. For the global minimum, fascin is in pink ($P = 0.69$); For the local minimum, fascin is in red ($P = 0.57$). **l**, Heatmap of the number of cross-bridges for filament pairs with different rotational shifts. Each filament has 14 protomers. **m**, Representative model renderings of the minima in the upper diagonal in **i**, showing fascin cross-bridges in the "down" pose. **n**, Representative model renderings of the minima in the lower diagonal in **i**, showing fascin crossbridges in the "up" pose. Blue boxes indicate filament $A$ rotation angles.

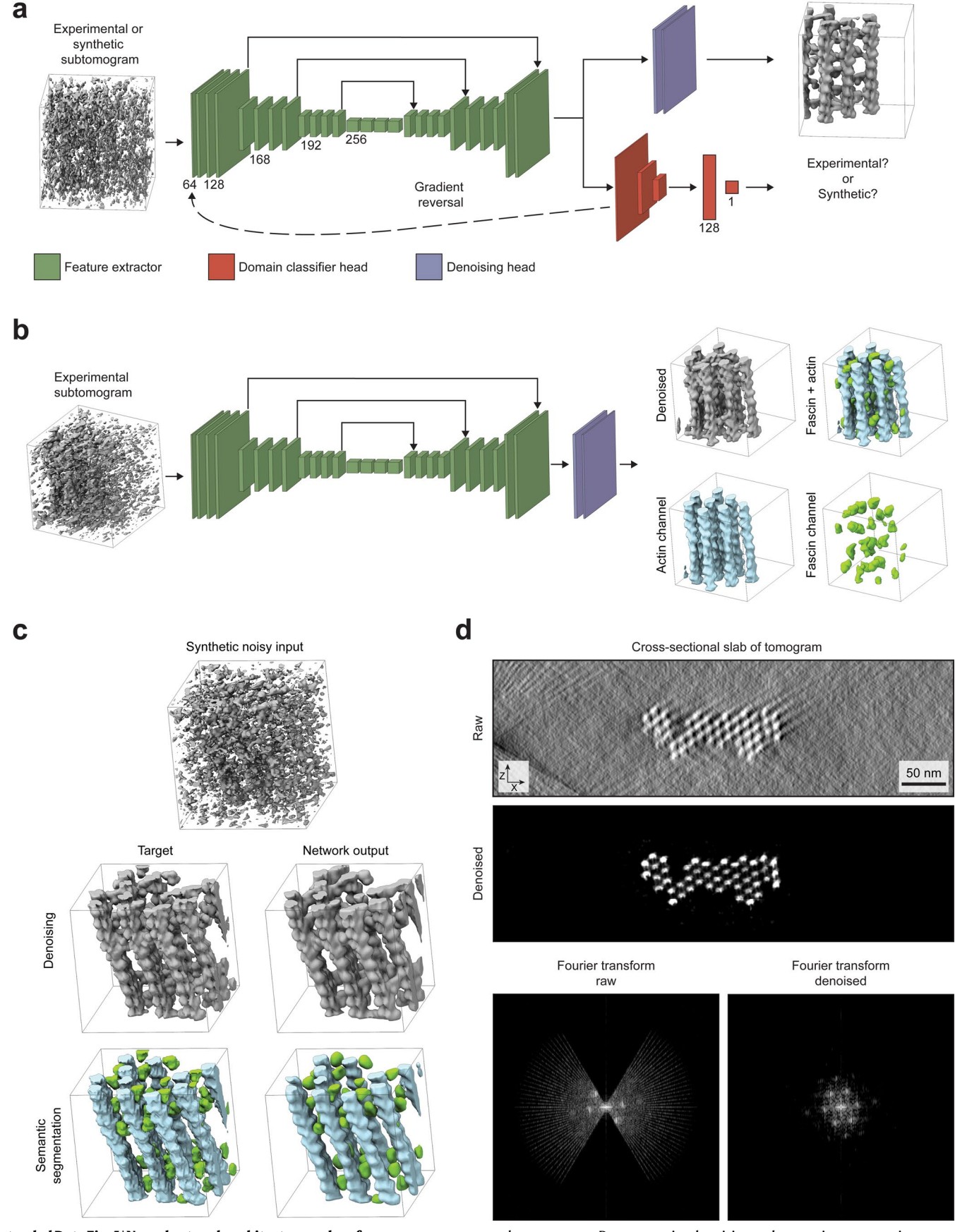

**Extended Data Fig. 5 | Neural network architecture and performance.**
**a**, Neural network architecture used for pretraining on synthetic data and
training on both synthetic and experimental subtomograms. **b**, Neural
network architecture and performance used for inference on experimental
subtomogram. **c**, Representative denoising and semantic segmentation
performance on synthetic, noisy subtomogram. **d**, Neural network denoising
performance on cross-sectional slab of experimental tomogram.

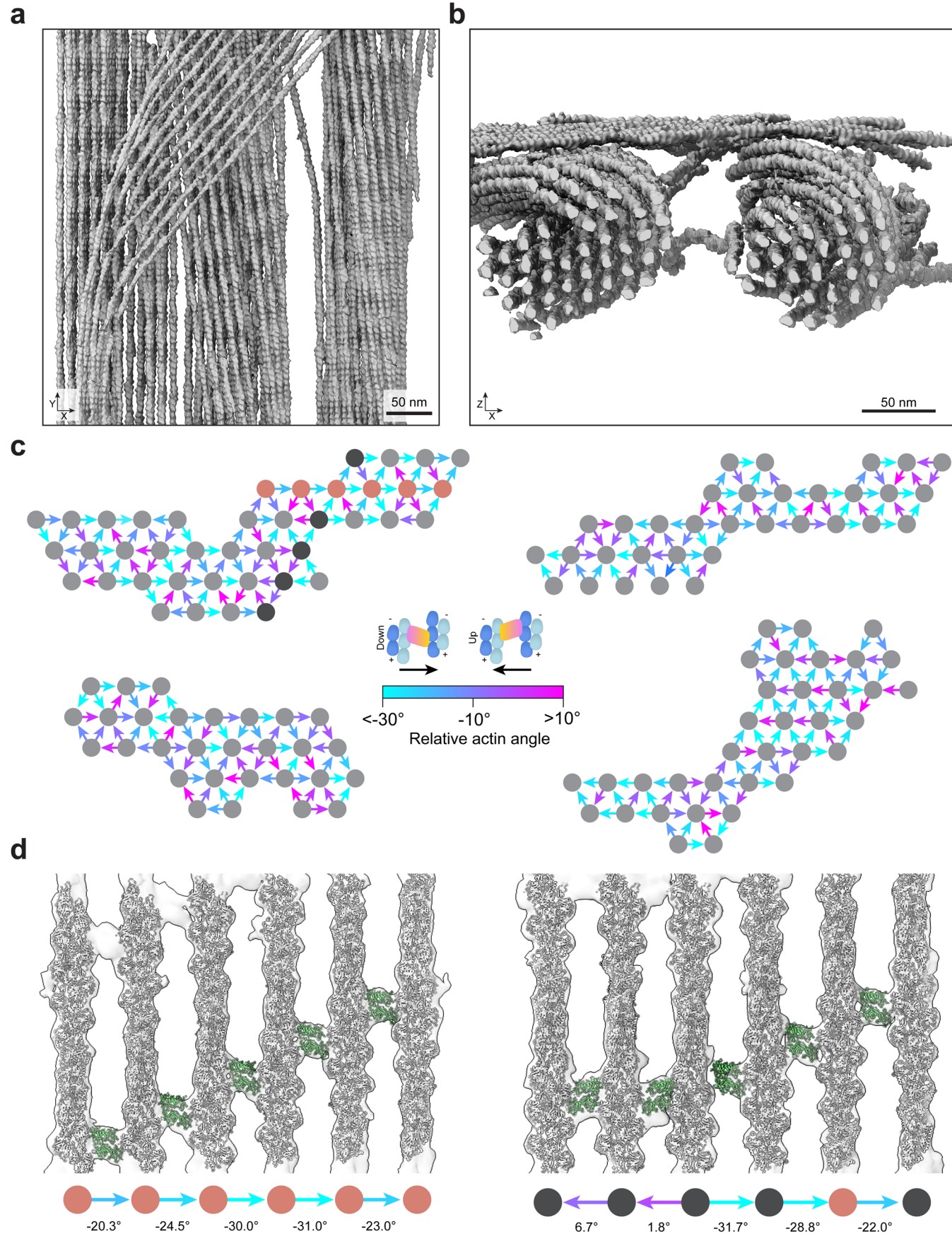

**Extended Data Fig. 6 | Additional structural analysis of fascin crosslinked bundles. a**, Top view of semantically segmented F-actin highlighting interconnected bundles. **b**, End-on view of bundles in **a**, highlighting supertwist along their respective longitudinal axes. **c**, End-on view schematics of filament rotational phase shifts and fascin poses of four additional bundles, analyzed as in Fig. 6b. Dark grey and gold filaments correspond to actin planes displayed in **d**. **d**, Side views of rigid-body docking models of two additional actin planes. Filament rotational phase offsets and fascin poses are indicated.

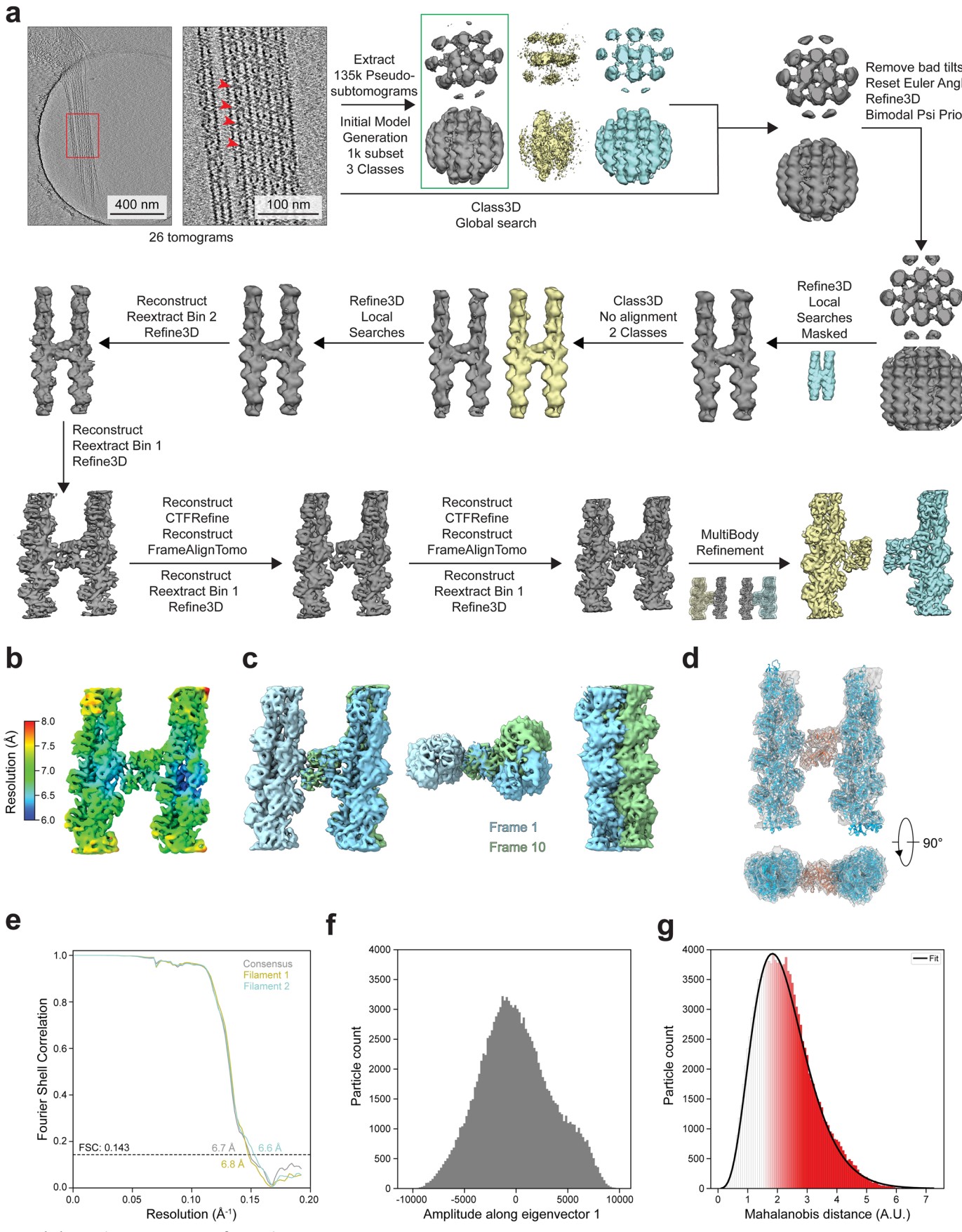

**Extended Data Fig. 7 | See next page for caption.**

**Extended Data Fig. 7 | Subtomogram averaging workflow and analyses.**
**a**, Subtomogram averaging data processing workflow. Top left: Slice of representative tomogram. Arrowheads indicate fascin cross-bridges. **b**, Local resolution map of consensus subtomogram average. **c**, Extreme frames (1 representing 5th percentile, and 10 representing 95th percentile) of interpolation along the first principal component of multi-body refinement. Similar inter-filament rotation is present as was observed in single particle analysis. **d**, Rigid body docking of the atomic model derived from single particle analysis into the consensus tomogram average, highlighting consistency between these independent analyses. Fascin is displayed in orange and F-actin in blue. **e**, FSC curves of the consensus (gray) and multi-body refinement reconstructions (yellow, blue). **f**, Distribution of amplitudes along eigenvector 1 of all subtomograms. **g**, Distribution of Mahalanobis distances of all subtomograms (n = 129,948). Fit represents gamma distribution ($R^2$ = 0.9951).

**Extended Data Table 1 | Filament helical parameters in the 460 Å bundle element**

| Filament ID | Consensus Rise (Å) (Relion 4.0) | Consensus Twist (°) (Relion 4.0) | Consensus Rise (Å) (hsearch_lorentz) | Consensus Twist (°) (hsearch_lorentz) | Local Twist (°) (Avg. ± Std. Dev.) | Axial Shift Relative to Filament 0 (Å) |
|---|---|---|---|---|---|---|
| 0 (Center) | 27.9 | -166.4 | 27.9 | -166.6 | -166.6 ± 0.7 | N/A |
| 1 | 27.9 | -166.6 | 27.9 | -166.5 | -166.4 ± 1.0 | -2.1 |
| 2 | 28.0 | -166.6 | 27.9 | -166.6 | -166.5 ± 1.4 | -1.0 |
| 3 | 28.0 | -166.3 | 27.9 | -166.4 | -166.6 ± 1.1 | 0.9 |
| 4 | 28.0 | -167.3 | 27.9 | -166.6 | -166.3 ± 2.5 | 2.1 |
| 5 | 27.9 | -166.7 | 27.8 | -166.1 | -166.1 ± 1.9 | 0.3 |
| 6 | 27.9 | -166.2 | 27.8 | -166.3 | -166.3 ± 1.2 | 1.5 |
| Overall Average | 27.9 | -166.6 | 27.9 | -166.4 | -166.4 | 1.3 |
| Std. Dev. | 0.05 | 0.4 | 0.05 | 0.2 | 0.2 | 0.7 |

**Extended Data Table 2 | Computational modeling probability function parameters**

| Function name | Distance assessed | Mean ($\mu$, nm) | Standard deviation ($\sigma$, nm) |
|---|---|---|---|
| P1 | $R95_i$ to $R95_{j+1}$ | 5.80 | 0.069 |
| P2 | $R95_i$ to $S350_{j+3}$ | 8.08 | 0.133 |
| P3 | $D24_{i+2}$ to $S350_{j+3}$ | 6.07 | 0.170 |
| P4 | $D24_{i+2}$ to $R95_{j+1}$ | 5.34 | 0.078 |
| P5 | Center of fiducial residues to plane spanning both filament axes | 11.287 | 3.0 |

## Extended Data Table 3 | Cryo-ET data collection, refinement and validation statistics

| | #1 Fascin crosslinked F-actin (Multibody: F-actin 1) | #2 Fascin crosslinked F-actin (Multibody: F-actin 2) | #3 Fascin crosslinked F-actin (composite map) (EMDB- 43372) |
|---|---|---|---|
| **Data collection and processing** | | | |
| Magnification | 26,000 | 26,000 | 26,000 |
| Voltage (kV) | 300 | 300 | 300 |
| Total electron exposure (e–/Å$^2$) | 109.17 | 109.17 | 109.17 |
| Electron dose per tilt image (e–/Å$^2$) | 2.66 | 2.66 | 2.66 |
| Energy filter slit width (eV) | 20 | 20 | 20 |
| Tilt range (°) | –60 to 60 | –60 to 60 | –60 to 60 |
| Acquisition Scheme | Dose-symmetric | Dose-symmetric | Dose-symmetric |
| Defocus (µm) | –4.0 | –4.0 | –4.0 |
| Pixel size (Å) | 2.6 | 2.6 | 2.6 |
| Symmetry imposed | C1 | C1 | C1 |
| Initial subtomograms (no.) | 134,733 | 134,733 | 134,733 |
| Final subtomograms (no.) | 129,948 | 129,948 | 129,948 |
| Map resolution (Å) | 6.8 | 6.8 | 6.7 |
| FSC threshold | 0.143 | 0.143 | 0.143 |
| Map resolution range (Å) | 6.1-12.6 | 6.1-12.6 | - |
| Map sharpening $B$ factor (Å$^2$) | -301.274 | -280.458 | -233.239 |

Rui Gong

# Reporting Summary

## Statistics

For all statistical analyses, confirm that the following items are present in the figure legend, table legend, main text, or Methods section.

| n/a | Confirmed | |
|---|---|---|
| ☐ | ☒ | The exact sample size (*n*) for each experimental group/condition, given as a discrete number and unit of measurement |
| ☒ | ☐ | A statement on whether measurements were taken from distinct samples or whether the same sample was measured repeatedly |
| ☐ | ☒ | The statistical test(s) used AND whether they are one- or two-sided<br>*Only common tests should be described solely by name; describe more complex techniques in the Methods section.* |
| ☒ | ☐ | A description of all covariates tested |
| ☐ | ☒ | A description of any assumptions or corrections, such as tests of normality and adjustment for multiple comparisons |
| ☐ | ☒ | A full description of the statistical parameters including central tendency (e.g. means) or other basic estimates (e.g. regression coefficient) AND variation (e.g. standard deviation) or associated estimates of uncertainty (e.g. confidence intervals) |
| ☐ | ☒ | For null hypothesis testing, the test statistic (e.g. *F*, *t*, *r*) with confidence intervals, effect sizes, degrees of freedom and *P* value noted<br>*Give P values as exact values whenever suitable.* |
| ☒ | ☐ | For Bayesian analysis, information on the choice of priors and Markov chain Monte Carlo settings |
| ☒ | ☐ | For hierarchical and complex designs, identification of the appropriate level for tests and full reporting of outcomes |
| ☒ | ☐ | Estimates of effect sizes (e.g. Cohen's *d*, Pearson's *r*), indicating how they were calculated |

*Our web collection on statistics for biologists contains articles on many of the points above.*

## Software and code

Policy information about availability of computer code

| Data collection | Cryo-EM and Cryo-ET data were collected with SerialEM v3.8. |
|---|---|

| Data analysis | For cryo-EM analysis, direct detector frame series were aligned with MotionCor2 v1.3.1, and CTF estimation was performed with CTFFIND4 v4.1.14. With the exception of 2D classifcation, ab initio reconstruction, and initial refinement performed with cryoSPARC v4.2.1, all other steps were performed with RELION4.0. |
|---|---|
| | For cryo-ET analysis, frame alignment and CTF estimation was performed with WARP v1.0.9. Tilt series alignment and reconstruction was performed with IMOD v4.11. Subtomogram averaging analysis was performed with RELION4.0. |
| | Custom code was used for particle picking and denoising / semantic segmentation of cryo-ET data, making use of libraries from EMAN2 v2.91 implemented in Python v3.7.9. This code was generated with the assistance of ChatGPT4.0. Custom code was also used for simulations, implemented in Matlab2022b. All custom code is available as open source at https://www.github.com/alushinlab/fascin |
| | Coot v0.9.2, ISOLDE v1.3, and Phenix v1.19.2 were used for atomic model building and refinement, while Molprobity v4.4 was used for validation. Chimera and ChimeraX v1.6.1 were used for molecular graphics, as well as structural analysis. DynDom was used to analyze domain motions.<br>The consensus rise and twist of actin filament was measured using both the relion_helix_toolbox in RELION 4.0 and the Iterative Helical Real Space Reconstruction (IHRSR) program hsearch_lorentz v1.5. |
| | Statistical analysis and plotting were conducted with GraphPad Prism 10. |

For manuscripts utilizing custom algorithms or software that are central to the research but not yet described in published literature, software must be made available to editors and reviewers. We strongly encourage code deposition in a community repository (e.g. GitHub). See the Nature Portfolio guidelines for submitting code & software for further information.

# Data

Policy information about availability of data

All manuscripts must include a data availability statement. This statement should provide the following information, where applicable:
- Accession codes, unique identifiers, or web links for publicly available datasets
- A description of any restrictions on data availability
- For clinical datasets or third party data, please ensure that the statement adheres to our policy

The cryo-EM density maps and atomic models generated in this study have been deposited in the Electron Microscopy Data Bank (EMDB) and Protein Data Bank (PDB): fascin bound filament 1 (EMDB: EMD-43364; PDB: 8VO5); fascin bound filament 2 (EMDB: EMD-43365; PDB: 8VO6); composite map of the fascin crossbridge (EMDB: EMD-43366; PDB: 8VO7); multibody binned reconstructions eigen_left: (EMDB: EMD-43367; PDB: 8VO8); eigen_middle: (EMDB: EMD-43368; PDB: 8VO9); eigen_right: (EMDB: EMD-43369; PDB: 8VOA); fascin crosslinked hexagonal bundle element with a box size of 460 Å (EMDB: EMD-43370); fascin crosslinked hexagonal bundle element with a box size of 740 Å (EMDB: EMD-43371); subtomogram average of fascin crossbridge (EMDB: EMD-43372). The trained neural networks for denoising and semantically segmenting micrographs and tomograms as well as the atomic models and volume maps used to generate synthetic datasets are available at Zenodo: https://doi.org/10.5281/zenodo.10456803. All other data are presented in the manuscript.

# Research involving human participants, their data, or biological material

Policy information about studies with human participants or human data. See also policy information about sex, gender (identity/presentation), and sexual orientation and race, ethnicity and racism.

| Reporting on sex and gender | N/A |
|---|---|
| Reporting on race, ethnicity, or other socially relevant groupings | N/A |
| Population characteristics | N/A |
| Recruitment | N/A |
| Ethics oversight | N/A |

Note that full information on the approval of the study protocol must also be provided in the manuscript.

# Field-specific reporting

Please select the one below that is the best fit for your research. If you are not sure, read the appropriate sections before making your selection.

☒ Life sciences  ☐ Behavioural & social sciences  ☐ Ecological, evolutionary & environmental sciences

For a reference copy of the document with all sections, see nature.com/documents/nr-reporting-summary-flat.pdf

# Life sciences study design

All studies must disclose on these points even when the disclosure is negative.

| | |
|---|---|
| Sample size | Sample sizes were not pre-determined. The amount of data collected were limited by the length of cryo-EM imaging sessions. Sufficient data were collected to achieve high-resolution reconstructions which addressed the scientific questions of the study. |
| Data exclusions | No data were excluded. |
| Replication | Highly similar reconstructions of the fascin crossbridge were obtained using single particle cryo-EM and subtomogram averaging, using independent methods and datasets. |
| Randomization | Particles were randomly assigned into half-datasets for resolution assessment by Fourier Shell Correlation analysis. |
| Blinding | No blinding was performed, as our study did not involve comparisons between different experimental conditions. Sorting into subpopulations was either performed via automated classification techniques, or by rigid-body docking analysis where orientation was determined through maximal crosscorrelation between the fascin crossbridge atomic model and denoised tomograms. |

# Reporting for specific materials, systems and methods

We require information from authors about some types of materials, experimental systems and methods used in many studies. Here, indicate whether each material, system or method listed is relevant to your study. If you are not sure if a list item applies to your research, read the appropriate section before selecting a response.

## Materials & experimental systems

| n/a | Involved in the study |
|---|---|
| ☒ | Antibodies |
| ☒ | Eukaryotic cell lines |
| ☒ | Palaeontology and archaeology |
| ☒ | Animals and other organisms |
| ☒ | Clinical data |
| ☒ | Dual use research of concern |
| ☒ | Plants |

## Methods

| n/a | Involved in the study |
|---|---|
| ☒ | ChIP-seq |
| ☒ | Flow cytometry |
| ☒ | MRI-based neuroimaging |

# Plants

| | |
|---|---|
| Seed stocks | N/A |
| Novel plant genotypes | N/A |
| Authentication | N/A |

