## [Peer Review File · Nature Structural & Molecular Biology]

Fascin structural plasticity mediates flexible actin bundle construction

Corresponding Author: Professor Gregory Alushin

Version 0:

Decision Letter:

16th May 2024

Dear Dr. Alushin,

Thank you again for submitting your manuscript "Fascin structural plasticity mediates flexible actin bundle construction". I apologize for the delay in responding, which resulted from the difficulty in obtaining suitable referee reports. Nevertheless, we now have comments (below) from the 3 reviewers who evaluated your paper. In light of those reports, we remain interested in your study and would like to see your response to the comments of the referees, in the form of a revised manuscript.

You will see that while reviewers appreciate the results, they raise several concerns which will need to be addressed in a revision. Specifically, we ask that you clarify the aspects of geometric analysis, as well as the details of the modelling, in line with comments of referees #2 and #3.

Please be sure to address/respond to all concerns of the referees in full in a point-by-point response and highlight all changes in the revised manuscript text file. If you have comments that are intended for editors only, please include those in a separate cover letter.

We expect to see your revised manuscript within 6 weeks. If you cannot send it within this time, please contact us to discuss an extension; we would still consider your revision, provided that no similar work has been accepted for publication at NSMB or published elsewhere.

Reporting Summary:

- that unprocessed scans are clearly labelled and match the gels and western blots presented in figures.
- that control panels for gels and western blots are appropriately described as loading on sample processing controls

-- all images in the paper are checked for duplication of panels and for splicing of gel lanes.

Please note that all key data shown in the main figures as cropped gels or blots should be presented in uncropped form, with molecular weight markers. These data can be aggregated into a single supplementary figure item. While these data can be displayed in a relatively informal style, they must refer back to the relevant figures. These data should be submitted with the final revision, as source data, prior to acceptance, but you may want to start putting it together at this point.

Data availability: this journal strongly supports public availability of data. All data used in accepted papers should be available via a public data repository, or alternatively, as Supplementary Information. If data can only be shared on request, please explain why in your Data Availability Statement, and also in the correspondence with your editor. Please note that for some data types, deposition in a public repository is mandatory - more information on our data deposition policies and available repositories can be found below:

<https://www.nature.com/nature-research/editorial-policies/reporting-standards#availability-of-data>

Link Redacted

Sincerely,

Katarzyna Ciazynska, PhD
(she/her)
Associate Editor
Nature Structural & Molecular Biology
<https://orcid.org/0000-0002-9899-2428>

Reviewers' Comments:

Reviewer #1:

Remarks to the Author:

Gong et al present an inspiring manuscript that uses state-of-the-art single particle and cryo-ET methods, along with simulation, to study actin/fascin bundles in vitro and in situ across multiple length scales. In doing so, they discover novel binding modes for fascin within actin bundles and demonstrate the flexibility of fascin binding to accommodate bundle plasticity. The manuscript was a pleasure to read with beautiful figures to illustrate the complex nature of the results. Their findings add exciting new clarity to the structural role of fascin in regulating actin bundling and would make a nice addition to your journal.

Reviewer #2:

Remarks to the Author:

The authors of this excellent and novel study used cryo-EM and sophisticated image analysis methods to observe a compact, horseshoe-shaped fascin, undergone an inter-domain rotations compared with its isolated structure, crosslinking a pair of actin filaments with an inter-filament distance of 12 nm. They find that G2 family of small molecules bind in a cleft in fascin and act as a molecular wedge, blocking deformations required to adopt actin-binding conformation. Elegant geometric analysis then leads to identifying two fundamental binding modes - up and down orientations of the fascin crossbridge. Next, the authors find that two actin-binding surfaces of fascin form completely divergent contacts with F-actin despite fascin's symmetry and that a considerable flexibility in fascin's binding configuration, positioning, rotational freedom and occupancy enable the assembly of hexagonal bundle elements with varying crosslinking geometries.

To test their understanding of the experimental results, the authors use a minimal computational model, in which actin filaments are sequentially added to a bundle with rotational phase shifts that maximize the probability of crosslinking. Consistent with the experimental results, the simulated assemblies featured substantial variability in the rotational phases of filaments and in the positioning of fascins, while maintaining the hexagonal bundle order. Analysis of denoised tomographs supported accumulated understanding of the crosslinked bundles' geometries. Lastly, analysis of fascin and filament deformations in thicker bundles, while not providing quantitative estimates for precise limit of fascin-crosslinked bundle size, suggests that there is an energetic barrier to thickening of the bundles.

I am absolutely not an expert on the experimental methods involved, and will leave the evaluation of respective aspects of the study to structural biologists, but to me the paper looks very carefully and clearly written. The findings are novel and important; the figures are great.

What I can evaluate is the model, which is accurate and of obvious utility for understanding the experimental results. My only critical comment is: in the lengthy description of the model in Methods, for a few first pages it looks like the model is based on purely geometric rules. Then, closer to the end, it becomes clear that effective energy functions and equilibrium thermodynamics are used to justify these rules. My advice is to revise by first spelling out the physical fundamentals, then show how the geometric rules were derived from physics, and then explaining the algorithm of growing bundles.

Reviewer #3:

Remarks to the Author:

This is an excellent paper describing many new structural and mechanistic insights into how fascin crosslinks actin filaments into bundles. This information will be extremely important in understanding cellular structures such as microvilli and stereocilia. The paper should be suitable for publication after minor revisions. I had a number of concerns and questions that the authors should be easily able to address.

p. 5) "interfilament distance of ~12 nm" But Jansen et al. (JBC, 2011) gave an interfilament distance of 8.1 nm, while Ishikawa et al. (J. Neurochem, 2003) gave 8-9 nm. Both of these earlier measurements are less than the diameter of an actin filament, raising questions about how this could be possible. But these previous observations should be mentioned.

p. 6) "Consistently, beta-trefoil 1 undergoes rotations with distinct directions and magnitudes in the G2 bound structure and our F-actin bridging structure versus the prebound state" This is quite unclear, given that three structures are being compared (G2, actin-bound and unbound).

p. 7, first line) "a shift of ~27 Å". But Fig. 1F has 28 Å. These should be consistent.

p. 10 and Fig. 4E) The rotational phase shifts are described. But to bring each peripheral filament into register with the central filament both rotations and axial translations may be needed. Is it really the case that when searching for both the axial translation is always 0 Å? This should be clarified.

p. 13, top paragraph, and Fig. 4E) This analysis can only be extended axially if actin filaments have exactly 13 subunits in 6 turns (a twist angle of -166.15 degrees, 2.1667 units/turn). Then a subunit 13 protomers above one in Fig. 4e will have exactly the same rotational phase shift. But the average twist of the filaments in these bundles is not reported, and it should be. Let us assume that it is -166.67 degrees (2.160 units/turn, consistent with observations in other papers and the 26.7 degree rotation described in this paper along a strand). Then a subunit 13 protomers above 1 in Fig. 4e would have a phase shift of -29.8 degrees. Similarly, a subunit 13 above 2 in Fig. 4e would have a phase shift of -0.9 degrees. This should be discussed.

p. 13) "absolute rotational shift of the central filament (which cannot be assigned in our experimental bundle element reconstructions)." This makes little sense to me as the central filament has no absolute rotational shift. Whatever its rotation is, that is taken as the reference for the surrounding 6 filaments. Perhaps I do not understand what is meant, and this needs to be clarified.

p. 19) There is another actin bundle that has been studied in some detail, the acrosomal process from *Limulus* sperm (Schmid et al., Nature, 2004). While the resolution of that particular study was much lower, the conclusions still appear sound that the "twist, tilt and rotation of actin-scrutin subunits deviate widely from a 'standard' F-actin filament." It would be informative to compare the fascin bundle with that structure.

Table S1) "Model resolution" should be replaced by "Map:Model resolution" since that is actually what is meant and what is being measured. The model resolution may actually be much higher, but what is being reported here is the agreement between the map and the model as a way to judge the resolution of the map.

Version 1:

Decision Letter:

Our ref: NSMB-A49011A

5th Jul 2024

Dear Dr. Alushin,

Thank you for submitting your revised manuscript "Fascin structural plasticity mediates flexible actin bundle construction" (NSMB-A49011A). It has now been seen by the original referees and their comments are below. The reviewers find that the paper has improved in revision, and therefore we'll be happy in principle to publish it in Nature Structural & Molecular Biology, pending minor revisions to satisfy the referees' final requests and to comply with our editorial and formatting guidelines.

We are now performing detailed checks on your paper and will send you a checklist detailing our editorial and formatting requirements in about 2-3 weeks. Please do not upload the final materials and make any revisions until you receive this additional information from us.

To facilitate our work at this stage, it is important that we have a copy of the main text as a word file. If you could please send along a word version of this file as soon as possible, we would greatly appreciate it; please make sure to copy the NSMB account (cc'ed above).

Sincerely,

Katarzyna Ciazynska, PhD
(she/her)
Associate Editor
Nature Structural & Molecular Biology
<https://orcid.org/0000-0002-9899-2428>

Reviewer #2 (Remarks to the Author):

I am satisfied with the revisions

Reviewer #2 (Remarks on code availability):

my impression is that the code is accurate and useful

Reviewer #3 (Remarks to the Author):

The authors have fully addressed my minor concerns, and the paper is suitable for publication.

Version 2:

Decision Letter:

17th Dec 2024

Dear Dr. Alushin,

We are now happy to accept your revised paper "Fascin structural plasticity mediates flexible actin bundle construction" for publication as an Article in Nature Structural & Molecular Biology.

Your paper will be published online soon after we receive proof corrections and will appear in print in the next available issue. You can find out your date of online publication by contacting the production team shortly after sending your proof corrections.

If you have not already done so, we strongly recommend that you upload the step-by-step protocols used in this manuscript to the Protocol Exchange. Protocol Exchange is an open online resource that allows researchers to share their detailed experimental know-how. All uploaded protocols are made freely available, assigned DOIs for ease of citation and fully searchable through nature.com. Protocols can be linked to any publications in which they are used and will be linked to from your article. You can also establish a dedicated page to collect all your lab Protocols. By uploading your Protocols to Protocol Exchange, you are enabling researchers to more readily reproduce or adapt the methodology you use, as well as

increasing the visibility of your protocols and papers. Upload your Protocols at www.nature.com/protocolexchange/. Further information can be found at www.nature.com/protocolexchange/about.

Please note that *Nature Structural & Molecular Biology* is a Transformative Journal (TJ). Authors may publish their research with us through the traditional subscription access route or make their paper immediately open access through payment of an article-processing charge (APC). Authors will not be required to make a final decision about access to their article until it has been accepted. [Find out more about Transformative Journals](https://www.springernature.com/gp/open-research/transformative-journals)

Sincerely,

Katarzyna Ciazynska, PhD
(she/her)
Senior Editor
Nature Structural & Molecular Biology
<https://orcid.org/0000-0002-9899-2428>

We thank the reviewers for their extremely positive comments and constructive feedback on our manuscript, which helped us to further improve it. We have provided a detailed response to each point below. We hope they will now find our paper acceptable for publication in NSMB. Please note that we have updated the figure / table referencing to conform with the journal's guidelines. Changes in the text are marked in purple.

Reviewer #1:

Remarks to the Author:

Gong et al present an inspiring manuscript that uses state-of-the-art single particle and cryo-ET methods, along with simulation, to study actin/fascin bundles in vitro and in situ across multiple length scales. In doing so, they discover novel binding modes for fascin within actin bundles and demonstrate the flexibility of fascin binding to accommodate bundle plasticity. The manuscript was a pleasure to read with beautiful figures to illustrate the complex nature of the results. Their findings add exciting new clarity to the structural role of fascin in regulating actin bundling and would make a nice addition to your journal.

We thank the reviewer for the very positive assessment of our work and support for its publication in NSMB.

Reviewer #2:

Remarks to the Author:

The authors of this excellent and novel study used cryo-EM and sophisticated image analysis methods to observe a compact, horseshoe-shaped fascin, undergone an inter-domain rotations compared with its isolated structure, crosslinking a pair of actin filaments with an inter-filament distance of 12 nm. They find that G2 family of small molecules bind in a cleft in fascin and act as a molecular wedge, blocking deformations required to adopt actin-binding conformation. Elegant geometric analysis then leads to identifying two fundamental binding modes - up and down orientations of the fascin crossbridge. Next, the authors find that two actin-binding surfaces of fascin form completely divergent contacts with F-actin despite fascin's symmetry and that a considerable flexibility in fascin's binding configuration, positioning, rotational freedom and occupancy enable the assembly of hexagonal bundle elements with varying crosslinking geometries.

To test their understanding of the experimental results, the authors use a minimal computational model, in which actin filaments are sequentially added to a bundle with rotational phase shifts that maximize the probability of crosslinking. Consistent with the experimental results, the simulated assemblies featured substantial variability in the rotational phases of filaments and in the positioning of fascins, while maintaining the hexagonal bundle order. Analysis of denoised tomographs supported accumulated

understanding of the crosslinked bundles' geometries. Lastly, analysis of fascin and filament deformations in thicker bundles, while not providing quantitative estimates for precise limit of fascin-crosslinked bundle size, suggests that there is an energetic barrier to thickening of the bundles.

I am absolutely not an expert on the experimental methods involved, and will leave the evaluation of respective aspects of the study to structural biologists, but to me the paper looks very carefully and clearly written. The findings are novel and important; the figures are great.

What I can evaluate is the model, which is accurate and of obvious utility for understanding the experimental results. My only critical comment is: in the lengthy description of the model in Methods, for a few first pages it looks like the model is based on purely geometric rules. Then, closer to the end, it becomes clear that effective energy functions and equilibrium thermodynamics are used to justify these rules. My advice is to revise by first spelling out the physical fundamentals, then show how the geometric rules were derived from physics, and then explaining the algorithm of growing bundles.

We thank the reviewer for the very positive feedback on our work and the valuable suggestions for improving our manuscript. As advised, we have reorganized the text and extended the introduction of the computational model in the Methods, building up to the algorithm from physical fundamentals (pp. 58-62). We agree this makes for a clearer presentation.

Reviewer #3:

Remarks to the Author:

This is an excellent paper describing many new structural and mechanistic insights into how fascin crosslinks actin filaments into bundles. This information will be extremely important in understanding cellular structures such as microvilli and stereocilia. The paper should be suitable for publication after minor revisions. I had a number of concerns and questions that the authors should be easily able to address.

We thank the reviewer for the very positive feedback on our work and particularly thoughtful suggestions for improving the paper.

p. 5) "interfilament distance of ~12 nm" But Jansen et al. (JBC, 2011) gave an interfilament distance of 8.1 nm, while Ishikawa et al. (J. Neurochem, 2003) gave 8-9 nm. Both of these earlier measurements are less than the diameter of an actin filament, raising questions about how this could be possible. But these previous observations should be mentioned.

We thank the reviewer for pointing out this discrepancy in literature. Most of the in vitro and in situ studies either before or after those two publications (Derosier, 1981; Yang, 2013; Aramaki, 2016; Atherton, 2022) consistently reported an inter-filament distance of ~ 12 nm. We believe the ~ 12 nm distance is the generally accepted parameter in the field. To avoid muddying the waters, we kindly prefer not to mention those two less accurate measurements in our main text.

p. 6) “Consistently, beta-trefoil 1 undergoes rotations with distinct directions and magnitudes in the G2 bound structure and our F-actin bridging structure versus the prebound state” This is quite unclear, given that three structures are being compared (G2, actin-bound and unbound).

In our analysis we compared two structures at a time. We found that beta-trefoil 1 undergoes rotations when one compares the unbound state to the F-actin bound state, as well as when one compares the G2 bound state to the F-actin bound state. These two rotations, however, have distinct directions and magnitudes. We have modified the text (p.6, second-to-last paragraph) to clarify this point.

p. 7, first line) “a shift of ~ 27 Å”. But Fig. 1F has 28 Å. These should be consistent.

We thank the reviewer for noticing this discrepancy. We have changed the text to “a shift of ~ 28 Å” for consistency.

p. 10 and Fig. 4E) The rotational phase shifts are described. But to bring each peripheral filament into register with the central filament both rotations and axial translations may be needed. Is it really the case that when searching for both the axial translation is always 0 Å? This should be clarified.

To address this point, we measured the axial shifts of all the six side filaments relative to the central one. We found that all the shifts are within the range of ± 2 Å, suggesting minimal contributions of inter-filament subunit axial shifts to the axial shifts of fascin. We have revised the text to clarify this point, and we report the axial shift measurements in new Supplementary Table 1, as well as the associated procedure in the Methods (p. 53, top section).

p. 13, top paragraph, and Fig. 4E) This analysis can only be extended axially if actin filaments have exactly 13 subunits in 6 turns (a twist angle of -166.15 degrees, 2.1667 units/turn). Then a subunit 13 protomers above one in Fig. 4e will have exactly the same rotational phase shift. But the average twist of the filaments in these bundles is not reported, and it should be. Let us assume that it is -166.67 degrees (2.160 units/turn, consistent with observations in other papers and the 26.7 degree rotation described in this paper along a strand). Then a subunit 13 protomers above 1 in Fig. 4e would have a phase shift of -29.8 degrees. Similarly, a subunit 13 above 2 in Fig. 4e would have a phase shift of -0.9 degrees. This should be discussed.

We thank the reviewer for the suggestion to measure and report helical twist, which was indeed an oversight in our initial submission. We have measured the consensus rise and twist of the filaments (as one does when performing Iterative Helical Real Space Refinement, IHRSR) using two different methods: the `relion_helix_toolbox` in Relion 4.0 and the program `hsearch_lorentz` developed by Ed Egelman. The former reported an average twist of -166.6° with a standard deviation of 0.4° , while the latter resulted in an average twist of -166.4° with a standard deviation of 0.2° . Both methods show that the twist of the filament in the bundle is similar to that of bare actin (-166.7°), suggesting fascin crosslinking has at most a minor effect on the twist of the filaments. The sub-nanometer resolution of the bundle also enabled us to accurately measure the local twist along each filament using the algorithm reported in Reynolds et al., Nature 2022. The local twists between individual subunits within the filaments are also quite uniform, with minor deviations similar to those observed in the asymmetric reconstruction of bare, straight ADP F-actin reported in Reynolds et al., Nature 2022. This aligns well with our three high resolution eigen-value based reconstructions derived from multi-body analysis, in which all six filaments reconstructed were nearly identical (Extended Data Fig. 3d). We summarized these measurements in Supplementary Table 1, and we have revised the main text (p. 11 bottom paragraph – p.12) and Methods (p. 53, top section) to include these analyses.

Accordingly, since the fascin bundle F-actin twist is similar to canonical F-actin, we completely agree with the reviewer's point that the rotational phase of the first protomer and the 14th protomer within each filament in the bundle are not identical. Therefore, there will be an overall rotation between each crossover layer as one proceeds axially along a bundle of approximately 3-5 degrees. However, since all the filaments in the hexagonal element have essentially uniform local twist that is highly similar to bare F-actin, when comparing two filaments, the relative phase difference between any index-matched pair of protomers from those filaments is effectively constant. Thus, the internal geometry of each transversal layer will be maintained, which is in effect what we have reconstructed through alignment and averaging in our bundle element reconstructions. Accordingly, the rotational phase differences reported in Fig. 4e represent the averaged relative phase difference between each filament pair, which should be uniform regardless of the protomer ID. We have clarified this in the main text. Nevertheless, as we note, this may decohere over longer length scales due to the moderate angular disorder of F-actin.

p. 13) “absolute rotational shift of the central filament (which cannot be assigned in our experimental bundle element reconstructions).” This makes little sense to me as the central filament has no absolute rotational shift. Whatever its rotation is, that is taken as the reference for the surrounding 6 filaments. Perhaps I do not understand what is meant, and this needs to be clarified.

We agree with the reviewer that this was confusing. Due to the finite filament lengths being simulated, the angular orientation of the central filament impacts the number and types of fascin bridges which are formed. This is not actually related to a limitation of experimental measurements. We have removed this statement and clarified the text

(p.14, bottom paragraph).

p. 19) There is another actin bundle that has been studied in some detail, the acrosomal process from *Limulus* sperm (Schmid et al., Nature, 2004). While the resolution of that particular study was much lower, the conclusions still appear sound that the “twist, tilt and rotation of actin-scruin subunits deviate widely from a 'standard' F-actin filament.” It would be informative to compare the fascin bundle with that structure.

We thank the reviewer for referring to earlier structural studies of the hexagonal bundle crosslinked by scruin, where filament structural plasticity was indeed observed. As suggested, we have compared and contrasted this system with fascin in the section of the Results which discusses twist (p. 11, bottom paragraph-p.12).

Table S1) “Model resolution” should be replaced by “Map:Model resolution” since that is actually what is meant and what is being measured. The model resolution may actually be much higher, but what is being reported here is the agreement between the map and the model as a way to judge the resolution of the map.

We have changed this to “Map:Model resolution” in Table 1.